# *Mycobacterium tuberculosis* progresses through two phases of latent infection in humans

Roberto Colangeli[1,8], Aditi Gupta[1,8], Solange Alves Vinhas[2,8], Uma Deepthi Chippada Venkata[1], Soyeon Kim[3], Courtney Grady[1], Edward C. Jones-López [4], Patricia Soteropoulos[5,6], Moisés Palaci[2], Patrícia Marques-Rodrigues[2], Padmini Salgame[1], Jerrold J. Ellner[1], Reynaldo Dietze[2,7] & David Alland [1✉]

Little is known about the physiology of latent *Mycobacterium tuberculosis* infection. We studied the mutational rates of 24 index tuberculosis (TB) cases and their latently infected household contacts who developed active TB up to 5.25 years later, as an indication of bacterial physiological state and possible generation times during latent TB infection in humans. Here we report that the rate of new mutations in the *M. tuberculosis* genome decline dramatically after two years of latent infection (two-sided p < 0.001, assuming an 18 h generation time equal to log phase *M. tuberculosis*, with latency period modeled as a continuous variable). Alternatively, assuming a fixed mutation rate, the generation time increases over the latency duration. Mutations indicative of oxidative stress do not increase with increasing latency duration suggesting a lack of host or bacterial derived mutational stress. These results suggest that *M. tuberculosis* enters a quiescent state during latency, decreasing the risk for mutational drug resistance and increasing generation time, but potentially increasing bacterial tolerance to drugs that target actively growing bacteria.

[1] Department of Medicine, Rutgers-New Jersey Medical School, Newark, NJ, USA. [2] Núcleo de Doenças Infecciosas, Universidade Federal do Espírito Santo (UFES), Vitória, Brazil. [3] Frontier Science Foundation, 1371 Beacon Street, Suite #203, Brookline, MA 02446, USA. [4] Division of Infectious Diseases, Department of Medicine. Keck School of Medicine of USC, University of Southern California Los Angeles, Los Angeles, CA, USA. [5] The Genomics Center, Rutgers-New Jersey Medical School, Newark, NJ, USA. [6] Department of Microbiology, Biochemistry, and Molecular Genetics, Rutgers-New Jersey Medical School, Newark, NJ, USA. [7] Global Health & Tropical Medicine, Instituto de Higiene e Medicina Tropical, Universidade Nova de Lisboa, Lisbon, Portugal. [8] These authors contributed equally: Roberto Colangeli, Aditi Gupta, Solange Alves Vinhas. ✉email: allandda@njms.rutgers.edu

Human infection with *Mycobacterium tuberculosis* produces an active form of disease known as tuberculosis (TB) or an asymptomatic form known as latent tuberculosis infection (LTBI)[1]. In LTBI, *M. tuberculosis* has been assumed to exist in a quiescent state, constrained by host immune responses within granulomas or other protected sites[2]. LTBI carries an ~5–15% risk of progressing to active TB within the first two years after infection. This is followed by an additional ~5% risk of developing active TB over the remaining lifetime of the infected host, usually due to waning immunity from disease or aging[2–4], although the true incidence of late TB recurrence during latency has recently been questioned[5]. Almost two billion of the world's population shows evidence of past *M. tuberculosis* infection, and if latently infected, may represent a major reservoir for new disease and a source of new *M. tuberculosis* transmission[6,7]. Recent studies have challenged a binary model of *M. tuberculosis* infection, which subdivides disease states into latent infection and active disease[8–11]. Instead, LTBI has been proposed to represent a spectrum ranging from immunological reactivity to a past cured infection, to sustained quiescent infection, to early low-level TB disease[5,8,12,13]. This model not only redefines latency, but also suggests that different types of preventive treatment might be more appropriate for different types of LTBI, assuming that the states of LTBI could be better identified in individual patients[10].

Understanding the physiological state of *M. tuberculosis* during latency is important for understanding the true impact of past *M. tuberculosis* infections on future TB, for identifying patients at high-risk of reactivation and for developing more efficient LTBI treatments. A critical part of these studies is a determination of whether *M. tuberculosis* is physiologically active, undergoing replication and mutagenesis during LTBI, or whether it exists in a quiescent state. In living organisms, genetic changes often occur at a steady rate per generation. For example, species with shorter generation times (i.e. the time required for bacteria to double in number, a measure of replication rate) tend to have higher rates of molecular evolution[14]. These observations have suggested that the number of mutations that occur over a defined period can be used as a "molecular clock" that provides an estimate of generation time (or replication rate) within a bacterial species[15,16]. Several studies have utilized molecular clocks to study the molecular evolution of *M. tuberculosis*[17–19]. Using cynomolgus macaques as a model for studying LTBI, Ford et al. showed that the mutation rates (i.e. the number of mutations that occur per generation time) occurring over a broad range of potential generation times in *M. tuberculosis* isolated from latent lesions and reactivated infections (cultured on average 41.7 and 68.1 weeks after infection, respectively) were comparable to those of *M. tuberculosis* isolated from patients with active pulmonary disease[20]. This work suggested that latent *M. tuberculosis* has generation times that are similar to that observed in active TB (or else mutation rates are much higher during latency). In contrast, Colangeli et al. performed a similar analysis comparing *M. tuberculosis* mutation rates in two TB patients with little to no latency period before symptomatic disease versus two patients with reactivated TB after two decades long latency period identified during a TB outbreak investigation in New Zealand. These results suggested that generation times are substantially longer during latency compared to active disease (or else mutation rates are far lower during latency)[21]. These conflicting studies illustrate how poorly the mutation rate, generation time, and the accompanying physiological and metabolic status of *M. tuberculosis* are understood during LTBI.

Here, we investigate the mutation rate and potential generation time of *M. tuberculosis* during LTBI by analyzing whole-genome *M. tuberculosis* sequences from 24 paired index TB cases and

their latently infected household contacts (HHCs) who were diagnosed with active TB 0–63 months after exposure. This first comprehensive study of *M. tuberculosis* mutation rates and generation times during latent infection shows that bacterial mutation rates appear to decline 1–2 years after transmission, coinciding with the clinical risk of developing active TB. These findings enable us to make predictions about treatment efficacy and the risk for developing bacterial drug resistance during LTBI treatment.

## Results

**Participant enrollment.** We measured the number of single nucleotide polymorphism (SNP) differences that existed between *M. tuberculosis* cultured from TB patients (index cases, IC) and the same *M. tuberculosis* strain cultured from secondary TB cases discovered among their HHCs. These results were then used to estimate the range of possible mutation rates and generation times of *M. tuberculosis* during LTBI. In a previously described HHC study performed in Vitória, Brazil between February 2008 and October 2013[22], we had identified 160 households that fit the case definition of containing a highly infectious case of acid-fast bacilli (AFB) smear-positive pulmonary TB, had at least three HHCs, and did not fulfill any exclusion criteria. Between the beginning of the HHC study in 2008 and the end of the study in 2015, 72 HHCs associated with 62 of these ICs were found to have developed active TB (Fig. 1). One IC–HHC pair was excluded due to lack of consent for TB genome analysis. A search though the local laboratory database located cultures from 43 of the remaining HHCs with TB, each linked to its known IC, resulting in 43 "TB pairs". To assess the genetic relatedness of each *M. tuberculosis* isolate within a TB pair, restriction fragment polymorphism (RFLP) analysis was performed on all isolates (Supplementary Fig. 1). Thirteen (30%) of the *M. tuberculosis* isolates within a TB pair differed by more than one band and were excluded from further analysis, due to an increased probability that the secondary case in the household had disease caused by a different *M. tuberculosis* strain than the one that infected the IC. In addition, the RFLP patterns of *M. tuberculosis* samples from two TB pairs matched that of *M. tuberculosis* isolates commonly circulating in the Vitória community (here, isolates that shared RFLP with >10 other isolates are considered as commonly circulating strains)[23]. These TB pairs were also excluded, as transmission between the IC and secondary HHC case could not be differentiated from the community circulating strain. Two TB pairs were excluded because the cultures of the secondary case could not be re-grown. The genomic DNA extracted from a confluent plate of each participant's sputum culture (too many colonies to count) of the remaining 26 TB pairs was purified and whole genome sequencing (WGS) was performed. One TB pair was excluded after sequencing because of poor genomic coverage and read depth. The average genomic coverage was 97.3% across all other isolates with an average read depth of 498× (Supplementary Table 1). One final TB pair was excluded after sequencing because phylogenetic analysis indicated that the two isolates did not share a most recent common ancestor (Supplementary Fig. 2). This analysis strongly suggested that the two individuals in this pair were infected with unrelated *M. tuberculosis* isolates, which was a pre-specified exclusion criterion. These exclusions left 24 TB pairs for subsequent analysis. Comparing these TB pair strains to a DNA fingerprinting study of 1357 *M. tuberculosis* isolates detected in the Vitória community between 1999 and 2015[24], revealed that 14 of the TB pairs included in our analysis had RFLP patterns that were unique to the TB pair and did not share a RFLP pattern with any other isolate studied in the community. The remaining 10 TB pairs

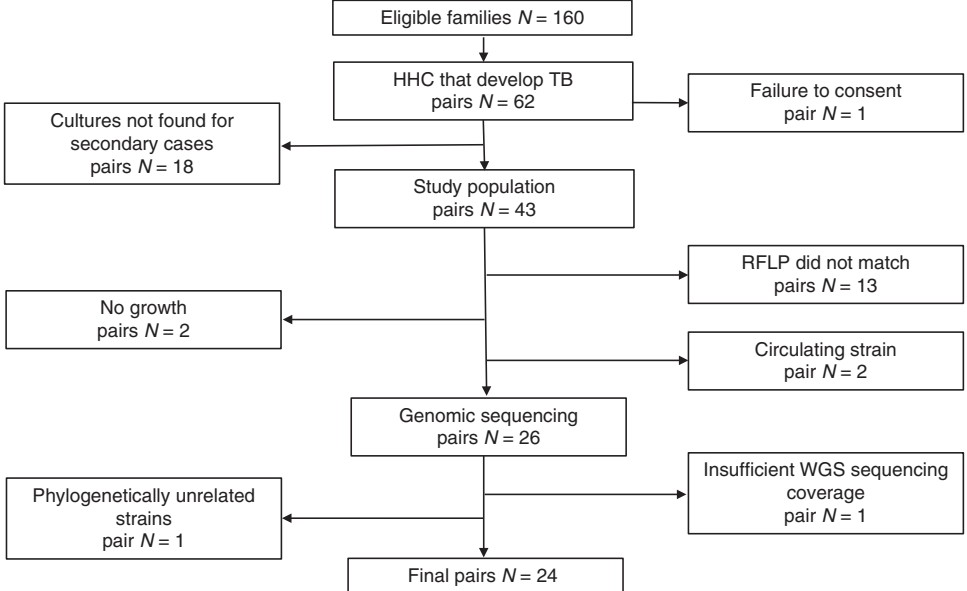

**Fig. 1 Sample inclusion chart.** IC index case, HHC household contact.

shared RFLP patterns with 3–7 other isolates in the community (medium-size cluster[24]). We also did not find evidence of mixed infection in any of the 24 TB pairs (Supplementary Fig. 3). All clinical isolates in this study were found to belong to *M. tuberculosis* lineage 4.

**Clinical characteristics.** Clinical characteristics of seven secondary cases were not available because these HHCs had not been selected for detailed evaluation in the parent study. The clinical characteristics of HHC cases for which clinical information was available are shown in Table 1. The HHC were mostly young adults with relatively low prevalence of co-morbidities, and moderate smoking and alcohol use. Most HHCs had evidence of BCG vaccination by the presence of a characteristic scar. None of the clinical characteristics of HHCs with early (0–2 years) versus late (>2 years) reactivation of TB disease were statistically distinguishable.

**The number of SNPs between IC and HHC isolates does not increase with duration of latency.** A phylogenetic tree was constructed for the 48 genomes in the 24 validated TB pairs. This analysis demonstrated that the *M. tuberculosis* isolates in the IC and the HHC within each TB pair differed from each other by an average of 2.25 SNPs (Supplementary Table 2), whereas the ICs and HHCs from unrelated TB pairs differed on average by 559.6 SNPs (Supplementary Fig. 4), supporting the epidemiological connection between each TB pair (Fig. 2). Next, we examined the number of SNPs that developed between each IC and the paired secondary TB case as a function of the time that separated the TB diagnosis in the pair. Figure 3a shows a scatterplot of the months between diagnoses and number of SNPs of the TB pairs with a loess smoothed curve. A Poisson regression model was fit to the difference in number of SNPs between IC–HHC pairs using months between the diagnoses as a linear term. Because of the small sample size and an estimated variance substantially higher than the mean, we used robust standard errors for statistical inference. Here, we assumed that the HHC was exposed to infection when the IC was diagnosed with active tuberculosis disease, and thus, the number of SNPs between the clinical isolates from IC and HHC reflected the approximate number of

**Table 1 Clinical characteristics of secondary *M. tuberculosis* cases from each TB pair according to time of reactivation.**

| Variable[a] | Early reactivation (0–2 yrs) | Late reactivation (>2 yrs) | p value[b] |
|---|---|---|---|
| Age (years) | 23.4 (n = 11) | 29.5 (n = 13) | 0.35 |
| Gender | | | 0.41 |
| Male | 7 | 5 | |
| Female | 4 | 8 | |
| Body mass index | 19.9 (n = 6) | 21.0 (n = 11) | 0.42 |
| BCG vaccine scar | | | 1.0 |
| Yes | 10 | 10 | |
| No | 1 | 2 | |
| NA | 0 | 1 | |
| HIV status (self-reported): | | | 1.0 |
| HIV Positive | 0 | 0 | |
| HIV Negative | 6 | 12 | |
| NA | 5 | 2 | |
| PPD max (mm) | 12.9 (n = 10) | 14.7 (n = 8) | 0.56 |
| Alcohol use | | | 1.0 |
| Yes | 1 | 1 | |
| No | 4 | 5 | |
| NA | 6 | 8 | |
| Smoking | | | 1.0 |
| Yes | 1 | 1 | |
| No | 4 | 5 | |
| NA | 6 | 8 | |
| Comorbidities[c] | | | 0.47 |
| Yes | 0 | 2 | |
| No | 5 | 5 | |
| NA | 6 | 7 | |

[a]Means are shown for numerical variables (age, body mass index, and PPD max), frequencies are shown for categorical variables (gender, BCG vaccine scar, HIV status, alcohol use, smoking, and co-morbidities). Not all data was available for each patient. Each variable value is indicated by the sample size used to derive the value.
[b]p Values comparing clinical characteristics in early vs. late reactivation are estimated using exact tests for categorical values (using available data only) and Wilcoxon rank sum tests for numerical variables.
[c]Comorbidities include: renal failure, diabetes, cancer, rheumatoid arthritis, COPD or gastric surgery.

bacterial mutations that occurred between the time of IC diagnosis and the HHC diagnosis. Our results show that the number of SNPs that differed between the *M. tuberculosis* genomic DNA of the ICs and their matched HHCs did not increase with the duration of the latency period (Fig. 3a, two-sided *p* = 0.90). This was unchanged with the exclusion of the outlier pair with SNP

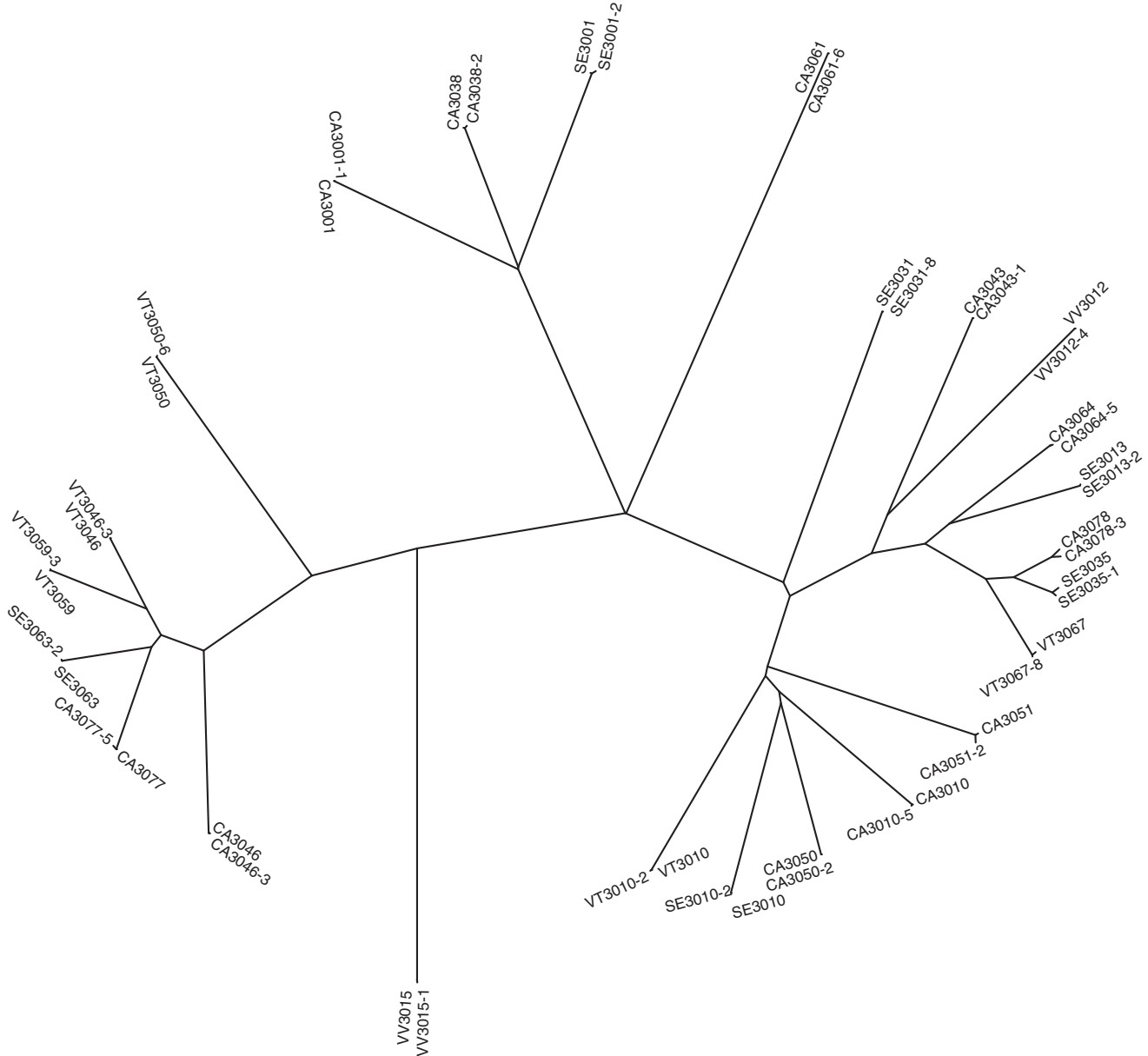

**Fig. 2 Phylogenetic relationship between *M. tuberculosis* isolates included in the study.** Neighbor joining tree constructed using pairwise SNP differences between all 48 isolates (24 IC and 24 HHC) is shown. The IC and HHC in each TB pair cluster together with the number in parenthesis denoting their SNP difference. The SNP differences between IC–HHC TB pairs ranged from 0 to 13. In contrast, any two unrelated *M. tuberculosis* isolates differed, on average, by 559.6 SNPs. Source data are provided as a Source Data file.

difference of 13. This result was maintained with an alternate SNP detection pipeline (Supplementary Table 3 and Supplementary Fig. 5). To test whether our results were influenced by HHCs who might possibly have acquired TB by exogenous re-infection from another TB patient in the community rather than the IC, we performed a subgroup analysis limited only to the 14 TB pairs that had a unique RFLP pattern not shared with any other isolate in the community (highlighted rows in Supplementary Table 2). This analysis showed similar findings to the full cohort ($p = 0.78$, Supplementary Fig. 6). To test whether the presence of sub-populations developing over time could influence the results, we studied the percent read support for each SNP across the genomes of our TB pairs over time. As shown in representative histograms (Supplementary Fig. 7), TB pairs that were diagnosed with TB even 63 months apart did not differ in intra-patient SNP diversity, which was very low in all samples.

**Mutation rate versus generation time**. The use of SNP differences as a replication clock assumes a fixed relationship between generation times and mutagenesis throughout the latency-disease spectrum. However, it is difficult to verify each of these measures during latency in humans due to a lack of active infection in LTBI that can produce culture positive sputum samples. Assuming a fixed 18 h generation time during latency, equal to that reported for log phase growth in *M. tuberculosis* cultures[25], we estimated the mutation rates (mutations/bp/generation) using a Poisson model for the number of mutations assuming a fixed common generation time for all TB pairs and used bp×generations as the denominator (by using its log as an offset in the model). We similarly fit a Poisson model to the SNP differences using the observed latency period as a continuous variable using the same offset, which showed a significant decrease in mutation rate with longer latency (two-sided $p < 0.001$, Fig. 3b, see Supplementary

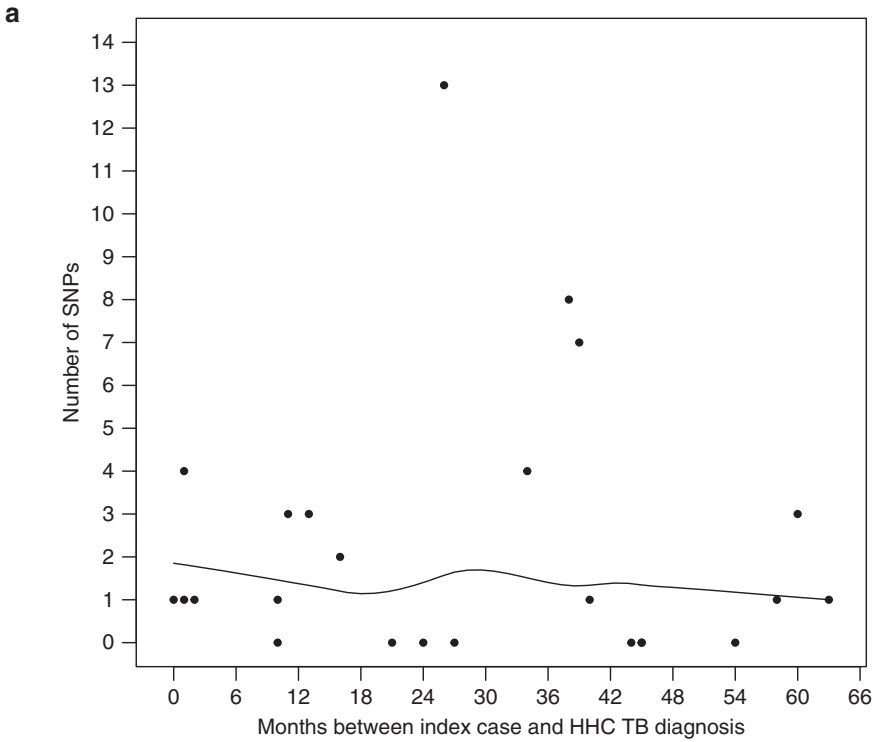

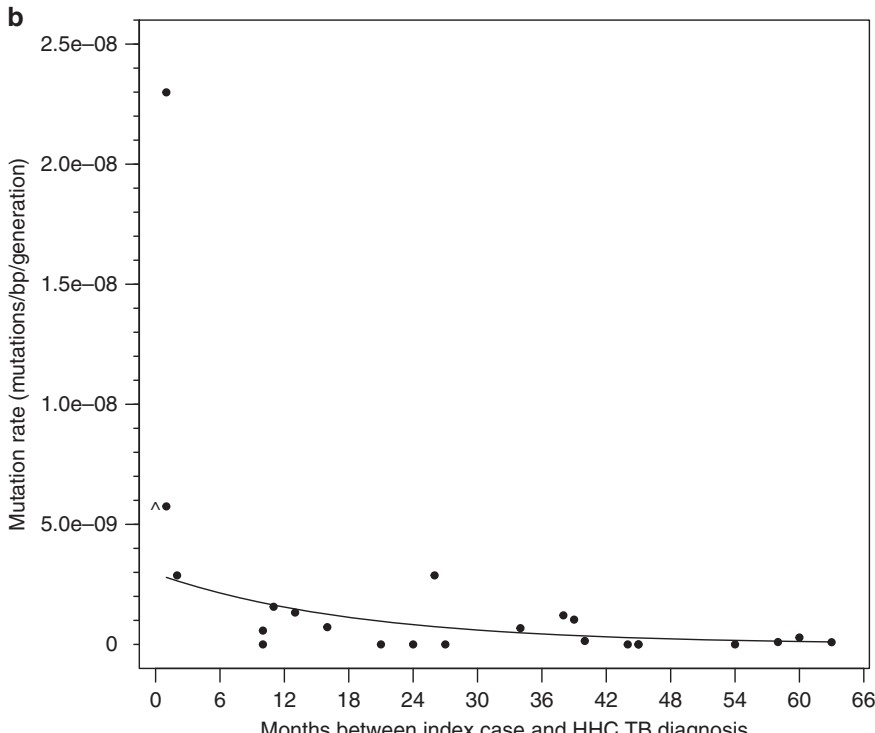

**Fig. 3 Mutation accumulation and mutation rate versus generation time during latency. a** Scatter plot showing the number of SNPs that differed between the $n = 24$ IC and the HHC isolate pairs (y-axis) as a function of the duration of *M. tuberculosis* latency (x-axis). The curve through the points was estimated using a loess local regression with a second degree polynomial and symmetric re-descending M estimator with Tukey's biweight function. **b** Mutation rate as a function of latency duration (months between IC and HHC TB diagnosis). The generation time is held constant at 18 h as seen in actively replicating *M. tuberculosis* in vitro. The smooth line shows the Poisson regression fit to the number of SNPs using $t$, the observed latency period in months, as the independent variable for each participant using an offset $N \times (t/g)$, where $N$ is 0.973×genome size and $t/g$ is the number of generations and $g$ is set to 18 h. The coefficient for $t$ was estimated to be −0.053 (95% confidence interval: −0.078, −0.029) and was significantly different from 0.0 (two-sided $p <$ 0.001). The point where the month between IC and HHC TB diagnoses was 0 was set to 1 month when fitting the model in order for it to contribute to the Poisson model and as is displayed as at month 0 as ˆ. Source data are provided as a Source Data file.

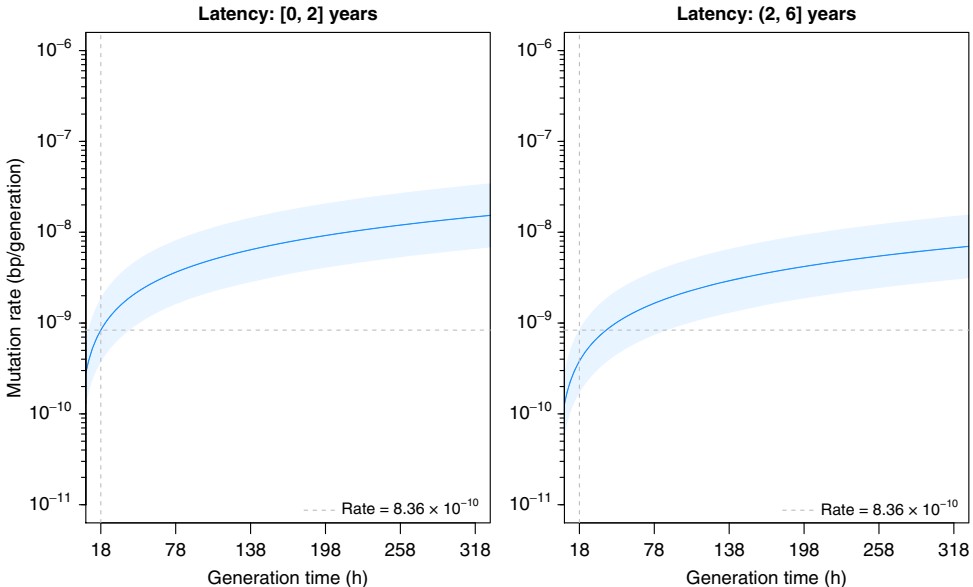

**Fig. 4 Changes in mutation rate during latency with varying generation time.** Mutation rate (mutations per (bp×generation)) is shown for generation times ranging from 18 to 320 h for IC–HHC pairs grouped by the number of years between IC diagnosis and reactivation of tuberculosis in the HHC. The dark blue line is obtained from the regressions and shows the estimated mutation rate for a given generation time (x-axis) and light blue regions show 95% confidence intervals. The first panel shows the relationship between mutation rate and generation time during early latency (reactivation in ≤2 years of IC diagnosis) based on $n = 11$ IC–HHC pairs. The second panel shows the relations between mutation rate and generation time for reactivation in >2–6 years after IC diagnosis based on $n = 13$ IC–HHC pairs. In both panels the gray dashed vertical line is at 18 h and the horizontal line indicates the mutation rate of $8.36 \times 10^{-10}$ mutations per (bp×generation) as seen in early latency (years 0–2, inclusive) with generation times held constant at 18 h. Source data are provided as a Source Data file.

Table 2 for raw data). Since the number of generations that elapsed during latency is inversely proportional to the generation time, the lack of SNP accumulation as a function of time in latency can be explained either by a decrease in bacterial mutation rate or a decline in bacterial replication leading to increased generation times during LTBI, or a combination of both. We therefore performed an analysis where both mutation rates (number of mutations per generation) and generation times could vary, calculating the mutation rate over a range of generation times[20] (18–318 h) (Fig. 4). When we binned the data into [0–2] and (2–6] years latency periods, we found that the mutation rates are substantially lower for the second period, estimated at $8.36 \times 10^{-10}$ [95% confidence interval (CI): $3.71 \times 10^{-10}$, $1.88 \times 10^{-9}$] mutations per (bp×generation) in ≤2 years of latency, to $3.81 \times 10^{-10}$ [95% CI: $1.70 \times 10^{-10}$, $8.53 \times 10^{-10}$] >2–6 year-long LTBI; however, the difference is not significant, two-sided $p = 0.18$. An analysis where we binned the data into three latency periods of [0–2], (2–4], and (4–6] years showed a statistically significant decline in the mutation rate over time (Supplementary Fig. 8).

To further understand how quickly bacteria enter a quiescent state post exposure, we simulated new SNP incidence during LTBI and found that majority of SNPs appear in a brief time window (assuming the bacterial generation time during LTBI to be similar to that during active TB). Thus, it is likely that majority of SNPs appear before the bacteria enters a quiescent state (i.e., soon after infection) or after the bacteria exits the quiescent state leading to reactivation of active TB (see Supplementary Methods).

**Oxidative stress during latency.** It has been suggested that the intracellular environment likely occupied by latent *M. tuberculosis* exposes the bacteria to excessive oxidative stress compared to active TB[20]. In particular, mutations due to cytosine deamination (cytosine to thymine and guanine to adenine) and 8-oxoguanine (guanine to thymine and cytosine to adenine) were reported to be abundant in macaques that reactivated to TB within 1–2 years

post infection[20]. We analyzed the proportion of oxidative stress mutations in all TB pairs compared to other types of mutations. Our results indicate that the proportion of oxidative stress mutations within the first 2 years of latency was indistinguishable (two-sided chi-square test $p$-value = 0.36) from that where the HHC developed disease 2–6 years after exposure to the IC (Fig. 5). This suggests that mutations attributed to oxidative stress, or similar mutations that could be the result of alterations in bacterial metabolism are not preferentially accumulating in *M. tuberculosis* during latency in humans.

**Detailed analysis of SNPs generated between TB pairs.** We performed an analysis of the SNPs that developed between all TB pairs to identify any patterns peculiar to our data set where all periods of latency where well defined. This analysis (Supplementary Table 2) revealed that 47.2% of SNPs were non-synonymous, 30.2% were synonymous and the remaining 22.6% SNPs were found in *M. tuberculosis* intergenic regions (Supplementary Fig. 9). All SNPs were found in different genes, suggesting that these SNPs likely did not have a functional role in latent infection (Supplementary Tables 4 and 5), although it is difficult to draw definitive conclusions about mutational distributions given the small numbers of mutations observed.

**Discussion**
Our results suggest that mutation rates are relatively high (and/or generation times are relatively low) up to 2 years after *M. tuberculosis* infection, which might more properly be considered an early incubation period or "early latency". *M. tuberculosis* then transitions into a physiological state of low mutation rate (or longer generation time) for the remaining duration of the latency period, which might be termed "late latency". Our results shed light on the long observed phenomenon that most cases of secondary TB occur within the first one to two years after infection[5], with the highest rate of reactivation occurring within the first

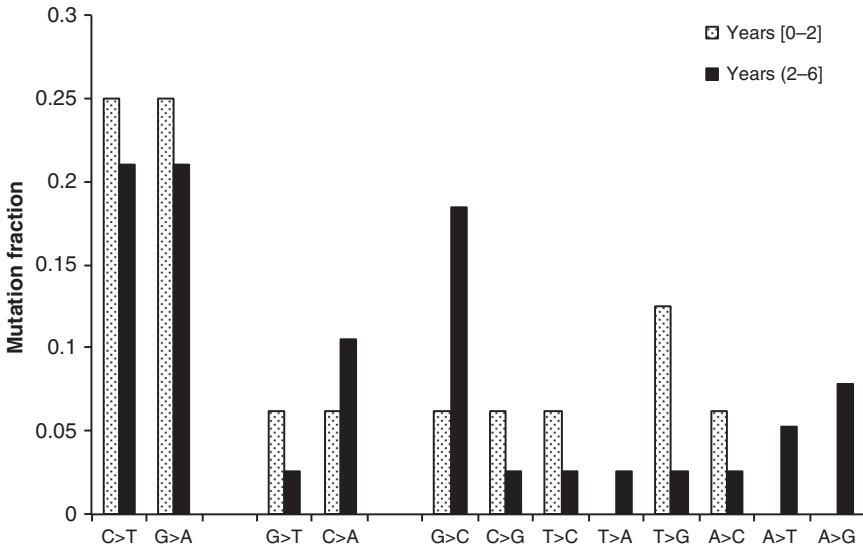

**Fig. 5 Oxidative stress during latency.** The proportion of mutations (*y*-axis) is shown for each mutation type (*x*-axis) for early (reactivation within 2 years, shaded bars) and late (reactivation after 2 years, black bars) latency. The incidence of mutations due to oxidative stress (cytosine deamination: C>T, G>A; and 8-oxoguanine: G>T, C>A) are indistinguishable in early and late latency. The number of mutations in early latency (years [0,2]) are: C>T: 4, G>A: 4, G>T: 1, C>A: 1, G>C: 1, C>G: 1, T>C: 1, T>A: 0, T>G: 2, A>C: 1, A>T: 0, A>G: 0. The number of mutations in late latency (years (2,6]) are: C>T: 8, G>A: 8, G>T: 1, C>A: 4, G>C: 7, C>G: 1, T>C: 1, T>A: 1, T>G: 1, A>C: 1, A>T: 2, A>G: 3. Source data are provided as a Source Data file.

year. This high-risk period approximates the period that we have termed early latency. On the other hand, TB reactivation rates from LTBI drop dramatically after the first two years of *M. tuberculosis* infection. This relatively protected period corresponds to the period that we have termed late latency. Our results also suggest that the optimal treatment for LTBI may vary depending on the time since infection is established, because many anti-tubercular drugs are only effective against actively replicating bacteria.

It is not possible to directly measure *M. tuberculosis* generation times in human latent infection. Instead, we performed an analysis that examined mutation rates over a wide range of possible generation times. We suggest however, with some assumptions, that it may be possible to use mutation rates to directly infer generation times during latent TB infection because we did not find any evidence that the distribution of mutations attributable to oxidative damage in early latency was significantly different from that in late latency. The use of mutation rates defined under laboratory conditions combined with measures of mutation accumulation over time in nature has been used to estimate replication rates in a wide variety of bacteria[16]. Assuming an 18 h generation time during latency, equal to that reported for log phase growth in *M. tuberculosis* cultures[25], our results predicted that mutation rates decreased with longer latency. Our binned analysis, which is known to result in reduced efficiency[26], comparing 0–2 years vs. >2 years LTBI suggested no difference in mutation rates. On the other hand, using mutation rate as a replication clock would lead to the conclusion that most of latency is characterized by a virtual absence of *M. tuberculosis* replication.

Our results may help to resolve the apparently contradictory conclusions reached by Ford et al.[20] and Colangeli et al.[21] by suggesting that both studies are likely correct. Ford et al. found that bacterial mutation rates per generation and generation times in cynomolgus macaques latently infected for <2 years were similar to that observed in active disease. In contrast, Colangeli et al. found that two human cases of latency lasting up to 20 years showed long bacterial generation times, especially after adjusting for the mutations measured in the two TB cases that developed

within 2 years after exposure. The contradiction between these two studies is resolved when it is appreciated that Ford et al. were likely studying early latency and Colangeli et al. were likely studying late latency.

One limitation of all HHC studies tracking TB transmission is the lack of absolute certainty that the index case was the source of bacteria that infected the HHC. We were fortunate in that TB transmission is relatively low in our Brazilian study community. Bayesian modeling of household vs. community TB transmission in the Vitória Brazil region estimated that only 1/5th of all HHC in this region are potentially infected from the community instead of the index patient[27]. Furthermore, we excluded TB pairs infected with *M. tuberculosis* strains that were commonly circulating in Vitória, and a sensitivity analysis including only TB pairs infected with unique strains produced identical results to the whole cohort. We could also not exclude the possibility that both the index and secondary case had been infected by a third source case. However, this is highly unlikely to have occurred, at least in the 14 TB pairs for which no other RFLP identical *M. tuberculosis* isolate was identified in the study community.

Our study is also limited by the nature of TB transmission and pathogenesis. TB is transmitted by aerosols, and the infecting dose is likely to be small, hence transmission bottlenecks may have caused some of the differences in the observed mutation rates between TB pairs. Indeed, our study did include one outlier that appeared to have a much higher mutation rate than any of the other TB pairs. However, a repeat analysis excluding this pair did not appreciably change our findings. Furthermore, this study was only able to detect mutations that were fixed in the *M. tuberculosis* population. Additional mutations that were even slightly deleterious might have been eliminated due to selection, leading to some underestimation of mutation rates. Thus, our study is most relevant to the population of fit bacteria that pose a risk for disease reactivation and not to these potential biological dead ends. We could not definitely conclude that the mutations which occur during early latency were not in fact a product of the post latency period, when all latent infections reactivate into active TB. However, the work by Ford et al. suggests that most of these mutations did indeed occur during an early latent period.

Nor can we be certain that the latency periods calculated for each TB pair are completely accurate, since the index case may have been infectious before it was detected as having TB disease in the household. Thus, it is possible that some of the latency periods in our study were longer than our estimates. However, neither of these possibilities would change the central findings of our study. In either scenario, our results show that mutations do not accumulate as a function of the period of latency lasting >2 years. Thus, most of latency is characterized by bacterial quiescence. Our study was performed in individuals with few co-morbidities, and none were known to have HIV-1 infection. It is possible that our findings would have been different if we had included individuals with risk factors that increase rates of TB reactivation. The high rates of LTBI reactivation known to occur in individuals with HIV-1 or other immunosuppressive conditions may be explained by ongoing replication of latently infected bacteria past the period that would normally define early latency. Finally, all the TB pairs in our study were infected with *M. tuberculosis* lineage 4. Although our study analysis benefits from this high degree of strain uniformity, it is possible that other TB lineages may exhibit different mutation rates × generation times during LTBI.

In summary, our study shows that measuring mutation rates per year of latency as well as potential bacterial generation times during LTBI shows that latency includes a prolonged period of low mutation rates or long generation times, likely indicating a quiescent state. We also show that mutations due to oxidative stress or alterations in bacterial metabolism do not confound this conclusion. Our findings suggest that modifying existing preventive therapy to accommodate the physiological transition of *M. tuberculosis* into a non-replicating state during prolonged latency is likely to improve the efficacy of LTBI treatment.

## Methods

**Human participants.** The study was approved by the Comitê de Ética em Pesquisa do Centro de Ciências da Saúde—Universidade Federal do Espírito Santo and the Comissão Nacional de Ética em Pesquisa under protocol number 14151, and the Institutional Review Boards of Boston University Medical Center and Rutgers University—New Jersey Medical School (formerly University of Medicine and Dentistry of New Jersey). We obtained written informed consent and assent in Portuguese in accordance with age-specific ethical guidelines of participating institutions.

**Study population and study samples.** We obtained *M. tuberculosis* cultures from participants enrolled in a HHC study performed at the Núcleo do Doenças Infecciosas (NDI) located in Vitória, the capital city of the State of Espírito Santo, Brazil. Participant enrollment and sample collection has been described[22]. In brief, the NDI has organized a network of 16 TB clinics in the metropolitan region of Vitória[28], which facilitated the identification of index TB cases as well as secondary TB cases in HHCs after initial ascertainment. All consecutive HIV-uninfected, pulmonary TB patients attending NDI clinics with a first episode of TB and a sputum with ≥2+ AFB smear and ≥3 HHCs who did not meet previously described exclusion criteria[22] were eligible for enrollment as IC. A HHC was defined using culturally adapted criteria of close contact[22]. All participating HHCs were enrolled within the first two weeks after the index TB patient had presented to the clinic. Co-prevalent active TB disease was excluded in all HHCs eligible for the current study by a detailed history and sputum examination at the time of the original HHC investigation. Chest X-rays examination was performed in all participants with symptoms suggestive of TB. To identify secondary cases of TB in the HHCs up to 6 years after identification of the index case, HHC study participants were matched with an extensive clinical and microbiological database "TB notes" of TB cases and *M. tuberculosis* cultures isolated in the greater Vitória region. This search identified 72 cases of TB in the HHCs. Seven of the HHC were found to have had a case of TB that predated the onset of TB in the "index case" that had led to the HHC investigation. In these seven cases, the first occurring TB case was defined as the index case and the second occurring TB case was re-defined as the secondary TB case.

**Culture of *M. tuberculosis* isolates and DNA isolation.** *M. tuberculosis* clinical isolates were cultured on Lowenstein Jensen (LJ) slants. *M. tuberculosis* H37Rv was grown from the same stock culture multiple times in Middlebrook 7H9 (BD, Sparks, MD) with 10% OADC supplement (BD, Sparks, MD) and 0.05% (w/v)

Tween-80 (Sigma, St. Louis, MO). Each strain was plated on Middlebrook 7H10 agar (BD Difco™) containing 0.02% glycerol and 10% OADC (BD Difco™ BBL™) and incubated at 37 °C. After substantial growth was observed, three loops of culture were scraped and suspended in SET (25% sucrose, 50 mM EDTA, 50 mM Tris pH 8.0) buffer containing 100 mg/mL lysozyme. DNA was then extracted using phenol–chloroform followed by purification using qiagen columns (Qiagen)[29]. DNA was further purified using QIAamp DNA mini kit.

**RFLP identification.** *M. tuberculosis* isolates, stored at −70 °C, were sub-cultured in Ogawa medium and all viable isolates were typed by the IS*6110* RFLP according to a standardized method[30]. The *M. tuberculosis* 14323 strain was used to compare patterns. The band patterns were analyzed with *BioNumerics* software version 7.6 (Applied Maths, Sint-Martens-Latem, Belgium). A dendrogram was constructed to elucidate the degree of similarity among the isolates. Unweighted pair group method with arithmetic mean (UPGMA) and the dice index (1.0% tolerance, 1.5% optimization) were used. Cluster analysis was performed using a cluster definition of two or more *M. tuberculosis* isolates RFLP hybridization patterns that were identical or that differed by only one band.

**Whole genome sequencing.** The Illumina libraries for clinical isolates were prepared using Nextera XT Index and Library prep and the DNA samples were multiplexed and sequenced on two lanes in Illumina HiSeq 2500 by the Genome Technology Center of the New York University. The Illumina libraries of in vitro cultures of laboratory strain H37Rv were prepared using the Illumina's Nextera Flex Library prep kit and Illumina's Nextera DNA CD Index kit (for adapters), and the normalized DNA samples were sequenced on NextSeq 550 Illumina system at The Genome Center of Rutgers-NJMS, Newark.

**SNP identification.** SNPs were identified using two SNP detection pipelines that are specific to *M. tuberculosis*: MTBseq[31] (version 1.0.4) and SNPTB[32] (version 1.0.0). In the MTBseq pipeline, raw reads were mapped to the reference genome (NCBI Accession: NC_000962.3) and SNPs that were supported by at least 75% of the reads, with a minimum of four reads in both forward and reverse directions containing the SNP, and with at least four reads having a minimum phred score of 20 were called. SNPs found in the repetitive regions of the genome or in genes associated with drug resistance were excluded. SNPs within a window of 12 bases were excluded as well. In the SNPTB pipeline, the raw reads were filtered using Trimmomatic[33] (version 0.36) and then aligned to the reference genome (NCBI Accession AL123456.3) using Bowtie 2[34] (version 2.2.6). Since each sample was sequenced twice on two separate Illumina lanes, the mapped reads were merged for each sample using SAMtools[35] (version 1.2) and high-confidence SNPs (QUAL score > 200) were identified using SAMtools (version 1.2)[35] and BCFtools[36] (version 1.2). All SNPs had a minimum coverage of 20 in all samples. SNPs found in the highly repetitive PE/PPE gene family were ignored.

**SNP validation.** The SNPTB pipeline was tested on four cultures of the laboratory reference strain of *M. tuberculosis*, H37Rv. Each culture was grown separately from the same stock and sequenced on the Illumina NextSeq platform following either 0, 1, 2, or 3 passages. The pipeline found the four cultures to be genetically identical (ignoring indels, and SNPs/indels in the PE/PPE genes). This ruled out laboratory culture, DNA extraction, or sequencing procedures as a source of SNPs.

**SNP annotation.** The SNPs were annotated as synonymous or non-synonymous by inserting the nucleotide change in the reference gene sequence and comparing the translations of the mutated and the reference gene sequences. SNPs were designated as intergenic if they lied in non-genic regions of the H37Rv genome. The SNPTB bioinformatics pipeline for identifying and annotating SNPs in *M. tuberculosis* (synonymous, non-synonymous, or intergenic) is available at https://github.com/aditi9783/SNPTB[32].

**Phylogenetic analysis.** A distance matrix was constructed by determining the pairwise SNP differences between all samples (24 IC and 24 HHC = 48 total samples). This distance matrix was then used to construct an unrooted phylogenetic tree using the neighbor-joining method as implemented in the PHYLIP package (version 3.696)[37,38]. The tree was drawn using the drawtree program of the PHYLIP package.

**Detecting mixed infections.** Sobkowiak et al. suggests that *M. tuberculosis* samples with >10 heterozygous sites are candidates for potential mixed infections[39]. We analyzed mapped reads for our 48 samples for presence of multiple polymorphisms at genomic positions with coverage >20 such that each SNP had at least 5% read support. None of the samples in our study had >10 heterozygous sites.

**Lineage detection.** The VCF files containing SNPs for each of the samples were submitted to the SNP-IT program to detect *M. tuberculosis* lineages for the samples[40].

**Mutation rate calculation**. The mutation rate per bp×generation was calculated as follows[20]. Briefly, the mutation rate per bp×generation is defined as

$$\mu = \frac{\sum_{i=1}^{n} m_i}{N \cdot \sum_{i=1}^{n} (t_i/g)} \quad (1)$$

where $\mu$ is the mutation rate, $m_i$ is the number of SNPs that differ in the $i$th IC–HHC pair out of $n$ pairs, $N$ is the genome size (since we had, on average, reads covering 97.3% of the *M. tuberculosis* genome, $N = 0.973 \times L$ where $L$ is reference genome size), $t_i$ is time since infection (in hours), and $g$ is generation time (in hours). Since $t_i$ was captured in months, when estimating the mutation rate per bp×generation the single HHC diagnosed with TB in the same month as the index TB case had the infection time set to 1 month and then converted to hours.

**Statistics**. Patient characteristics between HHC that reactivated TB early were compared to HHC that reactivated TB late using Wilcoxon rank sum test (for numerical values) and Fishers exact test (for categorical values) as implemented in the SciPy package (version 1.0.0)[41]. Two-sided *p*-values of these tests are reported in Table 1. The variation of mutation rate over a range of generation times was analyzed using a Poisson regression (as implemented in R using the glm function) and robust variances (using the sandwich package) to control for deviations from distributional assumptions. Poisson models were used to obtain mutation rates per (bp×generation) by using bp×generation an offset. A Poisson model was fit for each latency time period, 0–2 years and >2–6 years, with $n = 9$ and 13 IC–HHC pairs, respectively, using an intercept only model. By combining data from all pairs and fitting a model using a binary variable that indicated membership in the two latency time periods as the independent variable, we tested whether stress mutations were more abundant in late latency (reactivation after >2 years of latency). We also fit a Poisson model using the latency period as a continuous independent variable and tested whether the parameter was significantly different from 0.0. Testing of Poisson model parameters used a two-sided chi-square test using the robust variance.

**Reporting summary**. Further information on research design is available in the Nature Research Reporting Summary linked to this article.

## Data availability

Genomic data (raw sequence reads) from clinical isolates are available from NCBI Sequence Read Archive via BioProject accession number PRJNA475130. H37Rv (laboratory strain) genomic data is available from NCBI Sequence Read Archive via BioProject ID accession number PRJNA607763. The MTBseq pipeline used *M. tuberculosis* H37Rv reference genome at NCBI, Accession: NC_000962.3, for SNP detection. The SNPTB pipeline used *M. tuberculosis* H37Rv reference genome at NCBI, Accession: AL123456.3, for SNP detection. Source data are provided with this paper.

## Code availability

The bioinformatics pipeline MTBseq is available at https://github.com/ngs-fzb/MTBseq_source. The bioinformatics pipeline SNPTB is available at https://github.com/aditi9783/SNPTB. All other scripts for data analyses are available at https://github.com/aditi9783/TB_latency_scripts. All scripts are publicly available for use under the MIT license. Questions regarding python code usage should be addressed to the author A.G., and questions regarding R code usage should be addressed to author S.K.

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

## Acknowledgements

Research reported in this publication was supported by the National Institute of Allergy and Infectious Diseases of the National Institutes of Health under Award Numbers U19AI111276 and U01AI065663. Computational resources were provided by the Office of Advanced Research Computing (OARC) at Rutgers, The State University of New Jersey, under the National Institutes of Health Grant No. S10OD012346. The content of this article is solely the responsibility of the authors and does not necessarily represent the official views of the National Institutes of Health.

## Author contributions

R.C., A.G., P.Sa., R.D., E.C.J., J.J.E., and D.A. designed the study; R.C., S.A.V., U.D.C.V., and C.G. performed experiments; M.P., S.A.V., E.C.J., P.M.-R., and R.D. contributed patient-based and RFLP data from Brazil; P.So. performed the genomic sequencing, A.G. performed genomic analysis and developed SNPTB; A.G. and S.K. performed data analysis; R.C., A.G., S.K., and D.A. wrote the manuscript; all authors read and approved the final manuscript.

## Competing interests

The authors declare no competing interests.
