## [Peer Review File · Nature Communications]

Reviewers' comments:

Reviewer #1 (Remarks to the Author):

Determining the rates of *Mycobacterium tuberculosis* (MTB) cell division, chromosomal replication, and mutagenesis during human infection has been a major – but elusive – goal in the almost two decades in which whole-genome sequencing (WGS) technologies have been applied to clinical MTB isolates and related strains/species within the MTB Complex. A key problem confounding these studies is the lack of critical “controls”, which include precise knowledge about the time of infection (and, therefore, the duration of in-host (micro)evolution), the metabolic and replicative state of the bacilli during extended infection, and the impact of host-mediated kill on the mycobacterial population size. To address these and other limitations, researchers have to date used animal models of infection (both mouse and non-human primate), in some cases in combination with molecular reporters (such as a “clock plasmid”), and have applied mathematical models which allow large variations in cell division and/or chromosomal replication rates. In this submission, Alland and colleagues exploit well-defined household contact (HHC) pairs in a Brazilian cohort to estimate mutation rates by comparing single nucleotide polymorphisms (SNP) differences between index and contact isolates as a function of estimated time of latent infection (LTBI). Their analyses support a model in which mutation rates are generally low, with a clear transition to very low mutation rates (and, by implication, replication rates) as the duration of LTBI extends beyond one year. However, the following should be considered:

1. As noted in the introductory statement above, there are a number of terms which can be used to describe the cycle of bacterial cell division and the rate of acquisition of genetic mutations, including generation time, chromosomal replication rate, and rate of mutagenesis. Some of these terms are used in this article by Alland and colleagues and it would be very useful at the outset for the authors to define exactly how the terms are applied in this study and, consequently, precisely what is being measured (or inferred) biologically.
2. The rates calculated in this study are heavily dependent on the quality/validity of SNP calls from the WGS data. Recent work (e.g., see Andrews and colleagues <https://doi.org/10.1101/733642>) investigating the impact of genomic variant identification has highlighted the challenges (and inconsistencies) inherent in the use of different models for variant calling – notably, how the method applied can influence the result. In this study, very low SNP numbers carry very significant weight in determining mutation rates; therefore, it would be good to know whether the authors considered/tested the use of alternative methods for variant calling? The potential impact of the different approaches on their inferences of HHC pairs and/or transmission clusters (for example, the decision to exclude “commonly circulating strains”) also seems relevant to address explicitly.
3. The authors’ assumption that “HHC was exposed to infection when the IC was diagnosed with active TB disease” is critical to the analysis and all subsequent claims about the implications of their results, but might not be accepted broadly within the field. For this reason, much greater

justification is required for this assumption: do the authors honestly believe that transmission to the HHC occurred only at the time of index case presentation? If so, how can this assumption be reconciled with (i) their earlier contention that the “binary” model of latent infection and active disease is problematic and (ii) large numbers of published and ongoing studies indicating that individuals might be transmitting bacilli (and, as a corollary, that others might be infected) long before clinical presentation (of the transmitter), if at all?

4. Continuing from the above, the underlying microbiological assumption is that the dominant genotype (identified after culture and propagation in the laboratory) present in the index case is also the dominant strain (identified after culture and propagation in the laboratory) transmitted to the HHC. In the absence of defining SNPs – such as drug-resistance alleles – it seems problematic to conclude this linear connection. See for example <https://doi.org/10.1101/681502>

5. The authors have not addressed the potential impact of mycobacterial population size on the mutation rate calculations. That is, might the (assumed) very low bacillary numbers present during (extended) LTBI not impact the inferred mutation rate under those conditions – especially given the prediction that (even slightly) deleterious mutations might be eliminated from infecting (micro)populations owing to selection?

6. On P9, the authors claim “...these results show that the lack of SNP accumulation as a function of time in latency can be explained either by a decrease in bacterial mutation rate or a decline in bacterial replication leading to increased generation times during LTBI, or a combination of both.” This argument appears circular given that they only have a single definite measurement (SNP number) from which all other values (including mutation and replication rates) are inferred.

7. The authors use their results to propose an apparently arbitrary cut-off of ± 1 year LTBI as boundary between high and low replicative/mutational states. This is a bold claim (especially given the caution noted above that the absolute numbers of SNPs detected is extremely low), and risks adding further dogma to a field already labouring under “hard” definitions of clinical and other phenotypes with scant supporting evidence. In this context, their assumption (P11 L5) that the results “shed light” on the timing of secondary TB seems inconsistent with their own admission that none of the SNPs identified appears to be functional. Is the argument, then, simply that ongoing replication is more likely to manifest in secondary TB? If so, this seems intuitive, and not necessarily supported (or countered) by their data. This criticism might sound harsh, but my aim is to push the authors to consider whether the time-frame they claim hasn’t been applied to the data rather than revealed by the data. In other words, are they not themselves falling victim to a bias which already prevails?

8. The authors acknowledge (P12 bottom) that they cannot exclude mutations occurring in the post latency period. In addition to this issue, how have the authors addressed the concern that all mutations detected in clinical isolates are confounded by mutational events which occurred during propagation of a host-adapted clinical strain in laboratory media?

9. The authors contend that knowledge about the physiological state(s) of MTB during LTBI might inform preventive treatment approaches (P13 end). This seems an interesting notion, though it's difficult to assess given that they don't expand on this proposal. What do the authors envisage? And how might this be achieved practically?

Minor comments, typographical and other errors

1. The authors claim (P3 L3) that MTB enters a “dormant state”; to my knowledge, there is no evidence to support the capacity for true dormancy in MTB, so this term should be avoided. Perhaps “quiescent” or “non-replicating” would be preferable.
2. The Abstract (L10) concludes that the absence of mutations from oxidative stress implies “a lack of mutational stress from host effector mechanisms”. This conclusion is problematic in assuming that “effector mechanisms” are required for such mutations; on the contrary, there is ample evidence from other bacterial systems that oxidative stress arising during bacterial metabolism and, especially, transitions between metabolic states (e.g., aerobic to anoxic) might represent the dominant source of such mutations.
3. P3, bottom. Perhaps reversing the phrase to “replication and mutagenesis” would be preferable since the latter is a function of the former.
4. P4, bottom. The authors cite work from McKinney and colleagues in support of their (valid) contention that mutation rate, generation time, etc. are poorly understood in MTB infection; this citation seems out of place here since the thrust is on the fact that these are poorly “understood during LTBI”, yet the McKinney work focused on chronic infection in the mouse model – hardly an appropriate comparison, unless the authors are claiming that chronic mouse infection mimics LTBI, a contentious argument. Incidentally, it seems strange that the “clock plasmid” work of Sherman and colleagues (<https://doi.org/10.1038/nm.1915>) is not cited in the same context.
5. P8 “The use of SNP differences as a replication clock assumes constant rates of [replication, cell division, and] mutagenesis”

Reviewer #2 (Remarks to the Author):

This study examined genetic variation in 48 MTB culture isolates from the sputum of 48 patients with tuberculosis. The 48 isolates formed 24 pairs each consisting of an index case of active TB disease and a household contact case of someone whom developed active TB disease 6 months – 5.25 years after the index case and in whom TB transmission from the index case was supported by epidemiological and genetic data. The authors use the delay in time to development of active TB

disease for the household contacts as a proxy for duration of latent TB infection (before reactivation) and use genetic distance between the index – household contact pairs to measure in vivo mutation rates.

Their main finding is that the time to secondary TB disease is independent of the genetic distance between paired (index – household contact) isolates, supporting their hypothesis that most mutations arise in the initial stages of TB infection before MTB enters a non-replicating state in the later stages of latent infection. Generally, the data and analysis support this, however I have several suggestions with regards to the presentation of the analysis. Whole genome sequencing has previously been used to study transmission chains and mutations arising between hosts in several studies. The main takeaway from this work is outlined above and should be presented in a more concise manner.

MAJOR

Abstract

“Assuming an 18h generation time equal to that reported for log phase *M. tuberculosis*, mutation rates decreased from $1.48e-9$ mutations/bp/generation in ≤ 1 year of latency to $8.55e-11$ mutations/bp/generation in 5-6 year-long LTBI.”

> The mutation rate for the 5-6 year-long LTBI was determined from just a single data point (1 pair). As such this might be a poor estimate from what the actual mutation rate is for 5-6 year-long LTBI. I'd recommend generalizing a bit here or re-grouping into ≤ 1 year and 3-6 or 4-6 years of latency to increase the sample size for a better estimate of the mutation rates for longer LTBI. This applies to other points authors discuss this point in results and discussion.

> Note that the period of time after initial exposure (usually < 2 years see Behr M et al BMJ 2018) is traditionally called incubation period not latency or early latency. Active TB disease at this stage is usually called primary disease or disease progression not reactivation after latency. Typically latency is reserved to designate the period of time of skin test or IGRA positivity without symptoms beyond two years of disease exposure with no history of TB treatment. I acknowledge that there is likely a transition period or spectrum here, but nevertheless I suggest revisions along these lines to avoid confusion or misinterpretation of their results.

Results

“Subject enrollment”

> The authors appear to assume that isolates collected from the index case and the household contact case came from clonal infections. That is, neither subject within each pair was infected with more than 1 MTB strain. A mixed infection in either subject would heavily skew the results by affecting the number of SNPs called between each pair. Given the low sample size of 24 pairs, a mixed infection in just 1/48 subjects could influence the conclusions made by the authors. I recommend correcting for this by interrogating each sequenced sample for the presence of mixed strains; there are multiple ways to do this:

Sobkowiak, B., Glynn, J. R., Houben, R. M., Mallard, K., Phelan, J. E., Guerra-Assunção, J. A., ... & Parkhill, J. (2018). Identifying mixed *Mycobacterium tuberculosis* infections from whole genome sequence data. *BMC genomics*, 19(1), 613.

Wyllie, D. H., Robinson, E., Peto, T., Crook, D. W., Ajileye, A., Rathod, P., ... & Walker, A. S. (2018). Identifying mixed *Mycobacterium tuberculosis* infection and laboratory cross-contamination during *Mycobacterial* sequencing programs. *Journal of clinical microbiology*, 56(11), e00923-18.

“we assumed that the HHC was exposed to infection when the IC was diagnosed with active tuberculosis disease, and thus, the number of SNPs between the clinical isolates from IC and HHC reflected the approximate number of bacterial mutations that occurred between the time of IC diagnosis and the HHC diagnosis.”

> I have three comments on this (1) authors did not assess for mixed infections (as mentioned above) and (2) the authors are comparing isolates across transmission events and hence the transmission bottleneck is likely resulting in some of the differences in observed mutation rates between each pair. And it's possible the effect of this is not just “noise” there may be bias introduced. Given the data at hand, there's little one can do to get around this problem but I suggest at least mentioning it in the Results and making note of it in the Discussion as well. Finally (3) it is possible that in vitro culture of MTB has also resulted in new variant fixation related to in vitro growth, or to the loss of diversity in some samples (in vitro growth bottleneck). See Nimmo et al (<https://bmcgenomics.biomedcentral.com/articles/10.1186/s12864-019-5782-2>). This should also be discussed.

“Our results show that the number of SNPs that differed between the *M. tuberculosis* genomic DNA of the index case and its matched HHC did not increase with the time of the latency period (Fig. 3A).”

> This is the central finding of the study and analyses. However, it is important to note that each bin has a small sample size and one of the pairs has a genetic distance of 13 SNPs which is slightly high and, in some studies, would not be below the threshold for considering transmission between two subjects. The authors should consider a sensitivity analysis by removing this pair from the analysis to make sure their results do not change. Walker et. al. 2013 define 12 SNPs as the threshold for transmission and similarly to the authors, investigate genetic diversity between household contacts.

Walker, T. M., Ip, C. L., Harrell, R. H., Evans, J. T., Kapatai, G., Dedicoat, M. J., ... & Parkhill, J. (2013). Whole-genome sequencing to delineate *Mycobacterium tuberculosis* outbreaks: a retrospective observational study. *The Lancet infectious diseases*, 13(2), 137-146.

“our results predicted that mutation rates decreased from 1.48×10^{-9} mutations/bp/generation in ≤ 1 year of latency to 8.55×10^{-11} mutations/bp/generation in 5-6 year-long LTBI (linear regression test, $p=0.018$, Figure 4). We also studied the generation time predicted if mutation rates were kept constant at 1.48×10^{-9} mutations/bp/generation as seen in HHC that reactivated disease within the year. This analysis predicted that generation time increased from 18 hours in ≤ 1 year of latency to 312 hours in 5-6 year-long latency (linear regression test, $p=0.022$, Figure 4).”

> As mentioned for a statement in the abstract, the value of 8.55×10^{-11} mutations/bp/generation in 5-6 year-long LTBI is calculated based off of a single pair and is heavily dependent on a single data point.

> The linear regression tests in Figure 4 for both (A) the mutation rates holding the generation time fixed and (B) the generation times holding the mutation rate fixed assume normality of the errors in $\log(\text{mutation rate})$. This may not be a valid assumption as the sample size is very low (24). Since the expected number of mutations under neutral evolution follows a Poisson distribution, then a test better suited to assess significance of their findings is Poisson regression.

“All SNPs were found in different genes, suggesting that these SNPs likely did not have a functional role in latent infection”

> Given the small number of mutations, it is difficult to draw conclusions about the mutational distribution and the neutrality of the mutations. For example supplementary Tables 3 & 4 show

mutations occurring in *esxL* and *esxK* respectively, two genes that are located next to each other on the MTB chromosome and belong to the same pathway.

Discussion

“Our results suggest that the mutation rates are relatively high (and/or generation times are relatively low) up to approximately the first year after *M. tuberculosis* infection, which might more properly be termed “early latency”” AND

“In either scenario, our results show that mutations do not accumulate as a function of the period of latency lasting >1 year.”

> See comment above. The authors analyze data in slightly different time bins. Consistency would help with results interpretation. For example little data is given to support that mutation rates are relatively high “up to approximately the first year after MTB infection”. In the results data was binned into 0-2 years as “early latency”, none of the Figures directly support this statement.

Reviewer #3 (Remarks to the Author):

In the underlying manuscript the authors sought to determine the mutation rate of MTBC during latency in order to address the question if and how fast the bacilli replicate (and mutate) in an asymptomatic TB infection. This question is important for preventive therapy and the decision which drugs are going to be offered to contact persons. As latent TB patients don't excrete living bacteria, the authors compared specimens from active TB patients and their house hold contacts who developed active TB. Direct transmission was inferred by classical RFLP genotyping and confirmed by genome wide pairwise distances using whole genome sequencing. The main observation is that the mutation rate decrease or vice versa the generation time increases with time past infection. This leads to important consideration which drugs (targeting growing or quiescent bacteria) to be used for preventive treatment.

Major points:

I have two major points that needs to be considered for the conclusion.

The decline in mutation rate is mainly attributable to 4 pairs/datapoints reflecting 4-6 years of suspected latency (Figure 3A, line 180-188). The mutation rate is generally very low for MTBC and we also see in larger outbreaks over one or two decades almost identical strains isolated from the index case and very recently diagnosed patients infected by other contacts. So even in actively transmitting and replicating strains, probably by chance, one can observe cases with an apparently quiescent strain, that does not acquire any mutations over years. Further it is also discussed that only few bacilli are actually establishing an infection and that it is highly depended by chance which sub-population from an index patient is transmitted. Two contacts for instance can be infected by a clone that is 7 bp apart and one is identical to the majority strain population in the index patient.

To make the observation of a decreased mutation rate over time attributable to long-term TB latency with the low number of samples investigated here is not entirely convincing to me.

Another point is the assumption that an immediate infection occurred, once the index case was diagnosed. The authors mentioned in line 294 that each index case had more than 3 household contacts and probably some other close contacts. How likely is it that the index case is just the first patient diagnosed in a small transmission network (with few undetected links) and the short genetic distance measured here among four pairs (with 4-6 years expected latency) is just a matter of a missing link between the two enrolled patients?

The authors have sufficient genome wide coverage among their individual MTBC pairs, so a low-frequency analysis might give more insights into the real intra-patient diversity and variability/mutation rate. Do you find sub-populations (below 75%) that are more diverse in the "long-term latent" pairs?

Minor points:

Line 129: 99.9% coverage with how many reads at least?

FigS2 the case with matching RFLP patterns and unrelated genomes is most likely a DNA mix-up. Should be excluded with that reason, the figure assumes somehow that this could be a possible scenario. Overall phylogenetic lineages should be annotated to make clear if mutation rates are maybe lineage specific or if the observation is attributable to all MTBC strains (also in Figure 2).

Line 352: Could it be that the expected genome size N was larger in the calculation than the actual size of the analyzed genome? Considering that the authors excluded repetitive elements from the SNP analysis (approximately 10% of the genome), as well as SNPs with insufficient coverage, the actual genome size might be less than 90%.

Best regards

Reviewer #1 (Remarks to the Author):

Critique 1. As noted in the introductory statement above, there are a number of terms which can be used to describe the cycle of bacterial cell division and the rate of acquisition of genetic mutations, including generation time, chromosomal replication rate, and rate of mutagenesis. Some of these terms are used in this article by Alland and colleagues and it would be very useful at the outset for the authors to define exactly how the terms are applied in this study and, consequently, precisely what is being measured (or inferred) biologically.

Response:

We have now defined both generation time and mutation rate in the introduction. The term chromosomal replication rate is no longer used.

Critique 2. The rates calculated in this study are heavily dependent on the quality/validity of SNP calls from the WGS data. Recent work (e.g., see Andrews and colleagues <https://doi.org/10.1101/733642>) investigating the impact of genomic variant identification has highlighted the challenges (and inconsistencies) inherent in the use of different models for variant calling – notably, how the method applied can influence the result. In this study, very low SNP numbers carry very significant weight in determining mutation rates; therefore, it would be good to know whether the authors considered/tested the use of alternative methods for variant calling? The potential impact of the different approaches on their inferences of HHC pairs and/or transmission clusters (for example, the decision to exclude “commonly circulating strains”) also seems relevant to address explicitly.

Response:

We thank the reviewer for this suggestion. We used an alternate bioinformatics pipeline called SNPTB for detecting SNPs in *M. tuberculosis* samples that uses widely used read mapping (Bowtie 2) and SNP calling algorithms (SAMtools and BCFtools). This pipeline identified a higher SNP difference for all IC-HHC pairs but the overall trend of “no increase in SNP difference as duration of latency increased” was maintained. We have added this new data to the supplementary information (Table S3 and Figure S5) to reflect the SNPTB analysis.

We further analyzed data from only those 14 IC-HHC pairs that had unique RFLP patterns, thus increasing the confidence that the pair was infected by strains uncommon in the community. We find that this subgroup had the same overall trend of no increase in SNP differences as the duration of latency increased (Supplementary Figure S6).

Critique 3. The authors’ assumption that “HHC was exposed to infection when the IC was diagnosed with active TB disease” is critical to the analysis and all subsequent claims about the implications of their results, but might not be accepted broadly within the field. For this reason, much greater justification is required for this assumption: do the authors honestly believe that transmission to the HHC occurred only at the time of index case presentation? If so, how can this assumption be reconciled with (i) their earlier contention that the “binary” model of latent infection and active disease is problematic and (ii) large numbers of published and ongoing

studies indicating that individuals might be transmitting bacilli (and, as a corollary, that others might be infected) long before clinical presentation (of the transmitter), if at all?

Response: We agree with the reviewer that we cannot entirely rule out the possibility that the transmission event between the index case and the secondary case occurred before the time when the index case was diagnosed with active TB disease. However the possibility that our analysis included a numerous cases of miscalculation in the approximate transmission dates between the index and the secondary case is countered by our observation that mutation rate \times generation time curves (Fig. 3) are increasingly depressed as one moves from TB pairs that are separated by shorter times (0 – 2 years) versus intermediate times (2 – 4 years), versus the longest interval (4 – 6 years). Rather this trend strongly suggests that this observed latency interval is accurate. Furthermore, even in the unlikely event that we frequently miscalculated of the duration of latency occurred in our TB pairs, this mistake would only serve to increase the true duration of latency. Such a miscalculation would only further decrease our calculated mutation rate \times generation time. Thus, our key finding remains i.e. that latency >2 years includes a prolonged period of almost no mutations, and likely little replication.

The reviewer's comments have been incorporated in the discussion where we now state: "Nor can we be certain that the latency periods calculated for each TB pair are completely accurate, since the index case may have been infectious before it was detected as having TB disease. Thus, it is possible that some of the latency periods in our study were longer than our estimates. However, neither of these possibilities would change the central findings of our study. In either scenario, our results show that mutations do not accumulate as a function of the period of latency lasting >2 years. Thus, most of latency is characterized by bacterial quiescence."

Critique 4. Continuing from the above, the underlying microbiological assumption is that the dominant genotype (identified after culture and propagation in the laboratory) present in the index case is also the dominant strain (identified after culture and propagation in the laboratory) transmitted to the HHC. In the absence of defining SNPs – such as drug-resistance alleles – it seems problematic to conclude this linear connection. See for example <https://doi.org/10.1101/681502>

Response:

If the index case was infected with multiple genotypes that were different from that analyzed in the secondary case, one would have expected there would have been a higher than expected number of SNP differences between the TB pairs. In fact, this is what we expected to see when we started this study. The point is that there were virtually no SNP differences between TB pairs and the very small difference that was observed did not increase over time. The very small number of SNP differences across the entire >4 million base pair genome provides much stronger evidence of strain identity than a few drug resistance mutations (as suggested by the reviewer) can be expected to provide.

However, this reviewer's comment did make us question the degree to which the few SNPs we detected could have been created during the process of culturing each patient. To rule out

laboratory culture procedures as a source of these SNPs, we grew four separate cultures from the same glycerol stock of reference strain of *M. tuberculosis*, H37Rv, that were then sequenced on the Illumina platform. These four cultures were genetically identical (ignoring indels and SNPs in the variable PE/PPE regions that we have ignored in our LTBI analyses). We have now added this result in the “SNP validation” section of the methods.

Critique 5. The authors have not addressed the potential impact of mycobacterial population size on the mutation rate calculations. That is, might the (assumed) very low bacillary numbers present during (extended) LTBI not impact the inferred mutation rate under those conditions – especially given the prediction that (even slightly) deleterious mutations might be eliminated from infecting (micro)populations owing to selection?

Response: This is an interesting point. We have added to the discussion: “This study was only able to detect mutations that were fixed in the *M. tuberculosis* population. Additional mutations that were even slightly deleterious might have been eliminated due to selection, leading to some underestimation of mutation rates. Thus, our study is most relevant to the population of fit bacteria that pose a risk for disease reactivation and not to potential biological dead ends.”

Critique 6. On P9, the authors claim “...these results show that the lack of SNP accumulation as a function of time in latency can be explained either by a decrease in bacterial mutation rate or a decline in bacterial replication leading to increased generation times during LTBI, or a combination of both.” This argument appears circular given that they only have a single definite measurement (SNP number) from which all other values (including mutation and replication rates) and are inferred.

Response: There are two measures, SNP numbers AND the measured period between the index and the secondary cases. Furthermore, mutation rates and generation times are related as a longer generation time increase the calculated number of mutations per generation. These two measures and the relation between mutation rate and generation time allows us to define “mutation rate X generation time” estimates as Ford et al. performed in their Nature Genetics study. It is then possible to impute various plausible mutation rates (based on laboratory studies) and impute possible generation times.

Critique 7. The authors use their results to propose an apparently arbitrary cut-off of ± 1 year LTBI as boundary between high and low replicative/mutational states. This is a bold claim (especially given the caution noted above that the absolute numbers of SNPs detected is extremely low), and risks adding further dogma to a field already labouring under “hard” definitions of clinical and other phenotypes with scant supporting evidence. In this context, their assumption (P11 L5) that the results “shed light” on the timing of secondary TB seems inconsistent with their own admission that none of the SNPs identified appears to be functional. Is the argument, then, simply that ongoing replication is more likely to manifest in secondary TB? If so, this seems intuitive, and not necessarily supported (or countered) by their data. This criticism might sound harsh, but my aim is to push the authors to consider whether the time-frame they

claim hasn't been applied to the data rather than revealed by the data. In other words, are they not themselves falling victim to a bias which already prevails?

Response:

We apologize to the reviewers for the oversight of removing the analyses in a revision that supported this claim. We had simulated new SNP incidence in every six-month interval of LTBI and found that models that introduce majority of SNPs immediately after exposure or prior to reactivation generate SNP data that is indistinguishable from the observed SNP data. We have now included this simulation analysis in the Supplementary Information under "Supplementary Methods- Modeling new SNP incidence during progression of LTBI" and associated Figure S9.

Critique 8. The authors acknowledge (P12 bottom) that they cannot exclude mutations occurring in the post latency period. In addition to this issue, how have the authors addressed the concern that all mutations detected in clinical isolates are confounded by mutational events which occurred during propagation of a host-adapted clinical strain in laboratory media?

Response:

To rule out laboratory culture procedures as a source of SNPs, we grew four separate cultures from the same glycerol stock of reference strain of *M. tuberculosis*, H37Rv, that were then sequenced on the Illumina platform. These four cultures were genetically identical (ignoring indels and SNPs in the variable PE/PPE regions that we have ignored in our analyses). We have now added this result in the "SNP validation" section of the methods.

Critique 9. The authors contend that knowledge about the physiological state(s) of MTB during LTBI might inform preventive treatment approaches (P13 end). This seems an interesting notion, though it's difficult to assess given that they don't expand on this proposal. What do the authors envisage? And how might this be achieved practically?

Response: Our manuscript now includes the following statements in the Discussion: "Our results also suggest that the optimal treatment for LTBI may vary depending on the time since infection is established, because many anti-tubercular drugs are only effective against actively replicating bacteria." AND "Our findings suggest that modifying existing preventive therapy to accommodate the physiological transition of *M. tuberculosis* into a non-replicating state during prolonged latency is likely to improve the efficacy of LTBI treatment." We hope the reviewer agrees that these forward looking statements are adequate without going beyond that is supported by our results.

Critique Minor comments, typographical and other errors

10. The authors claim (P3 L3) that MTB enters a "dormant state"; to my knowledge, there is no evidence to support the capacity for true dormancy in MTB, so this term should be avoided. Perhaps "quiescent" or "non-replicating" would be preferable.

Response:

We have changed "dormant" to "quiescent".

Critique 11. The Abstract (L10) concludes that the absence of mutations from oxidative stress implies “a lack of mutational stress from host effector mechanisms”. This conclusion is problematic in assuming that “effector mechanisms” are required for such mutations; on the contrary, there is ample evidence from other bacterial systems that oxidative stress arising during bacterial metabolism and, especially, transitions between metabolic states (e.g., aerobic to anoxic) might represent the dominant source of such mutations.

Response: We agree and have changed this statement to “a lack of mutational stress arising from host effector mechanisms or alterations in bacterial metabolism.” We have made similar changes where relevant throughout the manuscript.

Critique 12. P3, bottom. Perhaps reversing the phrase to “replication and mutagenesis” would be preferable since the latter is a function of the former.

Response: Done.

Critique 13. P4, bottom. The authors cite work from McKinney and colleagues in support of their (valid) contention that mutation rate, generation time, etc. are poorly understood in MTB infection; this citation seems out of place here since the thrust is on the fact that these are poorly “understood during LTBI”, yet the McKinney work focused on chronic infection in the mouse model – hardly an appropriate comparison, unless the authors are claiming that chronic mouse infection mimics LTBI, a contentious argument. Incidentally, it seems strange that the “clock plasmid” work of Sherman and colleagues (<https://doi.org/10.1038/nm.1915>) is not cited in the same context.

Response:

We agree with the reviewer that the work by McKinney and colleagues referred to on page 4 is not as informative about LTBI as we implied. The clock plasmid work by Sherman and colleagues referred to by the reviewer is not informative for the same reason. We have deleted this section from the introduction, as suggested by the reviewer.

Critique 14. P8 “The use of SNP differences as a replication clock assumes constant rates of [replication, cell division, and] mutagenesis”

Response:

We agree and have added the “replication, cell division, and” section to the above sentence as suggested.

Reviewer #2 (Remarks to the Author):

Critique 1. The mutation rate for the 5-6 year-long LTBI was determined from just a single data point (1 pair). As such this is might be a poor estimate from what the actual mutation rate is for 5-6 year-long LTBI. I’d recommend generalizing a bit here or re-grouping into ≤ 1 year and 3-6 or 4-6 years of latency to increase the sample size for a better estimate of the mutation rates for

longer LTBI. This applies to other point authors discuss this point in results and discussion.

Response:

We thank the reviewer for the input. We have now grouped data as [0,2], (2-4], and (4-6] years of latency and amended the manuscript accordingly (see Figure 3). These three groupings have n=11, n=9, and n=4, respectively.

Critique 2. Note that the period of time after initial exposure (usually <2 years see Behr M et al BMJ 2018) is traditionally called incubation period not latency or early latency. Active TB disease at this stage is usually called primary disease or disease progression not reactivation after latency. Typically latency is reserved to designate the period of time of skin test or IGRA positivity without symptoms beyond two years of disease exposure with no history of TB treatment. I acknowledge that there is likely a transition period or spectrum here, but nevertheless I suggest revisions along these lines to avoid confusion or misinterpretation of their results.

Response:

There are many ways to define latency. Perhaps the most common is the operation one used by the World Health Organization, which defines latency as “a state of persistent immune response to stimulation by Mycobacterium tuberculosis antigens without evidence of clinically manifested active TB. Someone has latent TB if they are infected with the TB bacteria but do not have signs of active TB disease and do not feel ill.” This definition does not include a requirement for two years to have passed before the latent term applies, probably because the date of infection is not usually known for most LTBI patients. https://www.who.int/tb/areas-of-work/preventive-care/ltbi_faqs/en/. Similarly, Ford et al., refer to “latent infection” in their Nature Genetics paper on mutations rates during latency in cynomolgus macaques even though they sacrificed their animals between 281 and 299 days after infection. Thus, LTBI can have several meanings, and in most it is only defined by an immunological response. However, we agree that “early latency” is a term that we coined, and that others may prefer a different definition. We have therefore we have changed this sentence in the discussion to: “...up to 2 years after *M. tuberculosis* infection, which might more properly be termed “an early incubation period” or “early latency”, to include the reviewers equally valid term.

Critique 3. The authors appear to assume that isolates collected from the index case and the household contact case came from clonal infections. That is, neither subject within each pair was infected with more than 1 MTB strain. A mixed infection in either subject would heavily skew the results by affecting the number of SNPs called between each pair. Given the low sample size of 24 pairs, a mixed infection in just 1/48 subjects could influence the conclusions made by the authors. I recommend correcting for this by interrogating each sequenced sample for the presence of mixed strains; there are multiple ways to do this:

Sobkowiak, B., Glynn, J. R., Houben, R. M., Mallard, K., Phelan, J. E., Guerra-Assunção, J. A., ... & Parkhill, J. (2018). Identifying mixed Mycobacterium tuberculosis infections from whole genome sequence data. BMC genomics, 19(1), 613.

Wyllie, D. H., Robinson, E., Peto, T., Crook, D. W., Ajileye, A., Rathod, P., ... & Walker, A. S. (2018). Identifying mixed Mycobacterium tuberculosis infection and laboratory cross-contamination during Mycobacterial sequencing programs. *Journal of clinical microbiology*, 56(11), e00923-18.

Response:

We thank the reviewer for suggesting the possibility of mixed infection and for the references that outline methods to check for these. We followed the approach outlined in the 1st reference (Sobkowiak et al, BMC Genomic, 2018) that states that “only samples with >10 heterozygous sites will be considered potential mixed infections”. We did not find any of our 48 samples in the 24 TB pairs to satisfy this criterion (we analyzed genomic positions with >20X coverage for presence of multiple SNPs such that each SNP has at least 5% read support). This analysis is now included in the Results, Methods, and SI (Figure S3) of the revised manuscript.

Critique 4. “we assumed that the HHC was exposed to infection when the IC was diagnosed with active tuberculosis disease, and thus, the number of SNPs between the clinical isolates from IC and HHC reflected the approximate number of bacterial mutations that occurred between the time of IC diagnosis and the HHC diagnosis.”

I have three comments on this (1) authors did not assess for mixed infections (as mentioned above) and (2) the authors are comparing isolates across transmission events and hence the transmission bottleneck is likely resulting in some of the differences in observed mutation rates between each pair. And it's possible the effect of this is not just “noise” there may be bias introduced. Given the data at hand, there's little one can do to get around this problem but I suggest at least mentioning it in the Results and making note of it in the Discussion as well. Finally (3) it is possible that in vitro culture of MTB has also resulted in new variant fixation related to in vitro growth, or to the loss of diversity in some samples (in vitro growth bottleneck). See Nimmo et al (<https://bmcgenomics.biomedcentral.com/articles/10.1186/s12864-019-5782-2>). This should also be discussed.

Response:

- (1) Please see our answer to the reviewer comment #3 about the possibility of mixed infections, note that we have ruled out this possibility with a new analysis.
- (2) We have added the following to the discussion: “Our study is also limited by the nature of TB transmission and pathogenesis. TB is transmitted by aerosols, and the infecting dose is likely to be small, hence transmission bottlenecks may have caused some of the differences in the observed mutation rates between TB pairs.”
- (3) To rule out laboratory culture procedures as a source of SNPs, we grew four separate cultures from the same glycerol stock of reference strain of *M. tuberculosis*, H37Rv, that were then sequenced on the Illumina platform. These four cultures were genetically identical (ignoring indels and SNPs in the variable PE/PPE regions that we have ignored in our analyses). We have now added this result in the “SNP validation” section of the methods.

Critique 5. “Our results show that the number of SNPs that differed between the *M. tuberculosis* genomic DNA of the index case and its matched HHC did not increase with the time of

the latency period (Fig. 3A). This is the central finding of the study and analyses. However, it is important to note that each bin has a small sample size and one of the pairs has a genetic distance of 13 SNPs which is slightly high and, in some studies, would not be below the threshold for considering transmission between two subjects. The authors should consider a sensitivity analysis by removing this pair from the analysis to make sure their results do not change. Walker et al. 2013 define 12 SNPs as the threshold for transmission and similarly to the authors, investigate genetic diversity between household contacts.

Walker, T. M., Ip, C. L., Harrell, R. H., Evans, J. T., Kapatai, G., Dedicoat, M. J., ... & Parkhill, J. (2013). Whole-genome sequencing to delineate Mycobacterium tuberculosis outbreaks: a retrospective observational study. *The Lancet infectious diseases*, 13(2), 137-146.

Response:

We repeated the analysis excluding the outlying pair with a genetic distance of 13 SNPs. The rate for a generation time of 18 hours for the >2 to 4 year group decreases to 3.68×10^{-10} (95% CI: 2.50×10^{-10} , 9.07×10^{-10}) mutations/bp/generation. Comparisons of the mutation rates for a fixed generation time for three periods, ≤ 2 years, >2 to 4 years, >4 to 6 years gave two-sided $p=0.005$. All other results were similarly qualitatively unchanged.

Critique 6. “our results predicted that mutation rates decreased from 1.48×10^{-9} mutations/bp/generation in ≤ 1 year of latency to 8.55×10^{-11} mutations/bp/generation in 5-6 year-long LTBI (linear regression test, $p=0.018$, Figure 4). We also studied the generation time predicted if mutation rates were kept constant at 1.48×10^{-9} mutations/bp/generation as seen in HHC that reactivated disease within the year. This analysis predicted that generation time increased from 18 hours in ≤ 1 year of latency to 312 hours in 5-6 year-long latency (linear regression test, $p=0.022$, Figure 4).”

As mentioned for a statement in the abstract, the value of 8.55×10^{-11} mutations/bp/generation in 5-6 year-long LTBI is calculated based off of a single pair and is heavily dependent on a single data point.

Response:

The updated figure 4 shows a Poisson model fit to the data (assuming a constant generation time of 18 h) with observed latency period as a continuous variable, thus avoiding the binning of data. This model statistically supports a decline in bacterial mutation rate with an increase in duration of LTBI. Even in figure 3, we have now grouped data as [0,2], (2-4], and (4-6] years of latency and amended the manuscript accordingly, to avoid bins with low data points. These three groupings have $n=11$, $n=9$, and $n=4$, respectively.

Critique 7. The linear regression tests in Figure 4 for both (A) the mutation rates holding the generation time fixed and (B) the generation times holding the mutation rate fixed assume normality of the errors in $\log(\text{mutation rate})$. This may not be a valid assumption as the sample size is very low (24). Since the expected number of mutations under neutral evolution follows a

Poisson distribution, then a test better suited to assess significance of their findings is Poisson regression.

Response:

We agree with the reviewer and have fit a Poisson model to the data using the log(bp×number of generations) as an offset. Because the variance was larger than the mean, we also used robust variances for testing.

Critique 8. “All SNPs were found in different genes, suggesting that these SNPs likely did not have a functional role in latent infection”

Given the small number of mutations, it is difficult to draw conclusions about the mutational distribution and the neutrality of the mutations. For example supplementary Tables 3 & 4 show mutations occurring in *esxL* and *esxK* respectively, two genes that are located next to each other on the MTB chromosome and belong to the same pathway.

Response:

We have expanded the statement above as suggested by the reviewer in the underlined section: “All SNPs were found in different genes, suggesting that these SNPs likely did not have a functional role in latent infection (Supplementary Tables S4 and S5), although it is difficult to definitively draw conclusions about mutational distributions given the small numbers of mutations observed”.

Critique 9. “Our results suggest that the mutation rates are relatively high (and/or generation times are relatively low) up to approximately the first year after *M. tuberculosis* infection, which might more properly be termed “early latency”” AND “In either scenario, our results show that mutations do not accumulate as a function of the period of latency lasting >1 year.”

See comment above. The authors analyze data in slightly different time bins. Consistency would help with results interpretation. For example little data is given to support that mutation rates are relatively high “up to approximately the first year after MTB infection”. In the results data was binned into 0-2 years as “early latency”, none of the Figures directly support this statement.

Response:

We agree that we should have more consistency in the manuscript. We have now harmonized the entire manuscript around the three analytic periods 0-2, 2-4 and 5-6 years. Figure 4 treats duration of latency as a continuous variable instead of binned analytic periods.

Reviewer #3

Critique 1. The decline in mutation rate is mainly attributable to 4 pairs/datapoints reflecting 4-6 years of suspected latency (Figure 3A, line 180-188). The mutation rate is generally very low for MTBC and we also see in larger outbreaks over one or two decades almost identical strains isolated from the index case and very recently diagnosed patients infected by other contacts. So

even in actively transmitting and replicating strains, probably by chance, one can observe cases with an apparently quiescent strain, that does not acquire any mutations over years. Further it is also discussed that only few bacilli are actually establishing an infection and that it is highly depended by chance which sub-population from an index patient is transmitted. Two contacts for instance can be infected by a clone that is 7 bp apart and one is identical to the majority strain population in the index patient. To make the observation of a decreased mutation rate over time attributable to long-term TB latency with the low number of samples investigated here is not entirely convincing to me.

Response:

In response to this review as well as comment by other reviewers, we have binned all of our analyses into three time periods 0-2, 2-4 and 5- 6 to increase the numbers in each period. We agree that some of the variation in SNP differences can be due to chance infection of a single clone (the change made in the discussion related to this issue is also discussed in our response to reviewer #2), however, what is remarkable is the overall consistency in SNP differences among all of the 24 TB pairs. This constitutes that largest study of its kind (Colangeli et al., only had two and two late TB pairs), and the statistical tests confirm the significance of these observations.

Critique 2. Another point is the assumption that an immediate infection occurred, once the index case was diagnosed. The authors mentioned in line 294 that each index case had more than 3 household contacts and probably some other close contacts. How likely is it that the index case is just the first patient diagnosed in a small transmission network (with few undetected links) and the short genetic distance measured here among four pairs (with 4-6 years expected latency) is just a matter of a missing link between the two enrolled patients?

Response:

Please see our response to reviewer number one to this same question. In particular please see our sub analyses on 14 TB pairs where the strain as defined by RFLP analysis was only present in the TB pair in our study and not in any other patients in the community, including others in the household.

Critique 3. The authors have sufficient genome wide coverage among their individual MTBC pairs, so a low-frequency analysis might give more insights into the real intra-patient diversity and variability/mutation rate. Do you find sub-populations (below 75%) that are more diverse in the “long-term latent” pairs?

Response:

The histograms of % read support for SNPs do not show more diversity in the long-term latent pairs. We have included representative histograms for the pair SE3013 and SE3013-2 where the IC and HHC were diagnosed 63 months apart (Supplementary Figure S8). This data suggests that the bacteria enters a truly quiescent state during LTBI, with no substantial increase in genomic diversity in reactivated patients even 5-6 years after exposure. We have now included these data in the results and new Supplementary Figure S7.

Critique 4. Line 129: 99.9% coverage with how many reads at least?

Response:

The number of mapped reads (post quality control) ranged from 9.65 million to 20.4 million reads, with an average of 16.65 million mapped reads in each sample. We have now added this information to the legend of Table S1.

Critique 5. FigS2 the case with matching RFLP patterns and unrelated genomes is most likely a DNA mix-up. Should be excluded with that reason, the figure assumes somehow that this could be a possible scenario.

Response:

The figure legend clearly states that this pair was excluded. The point of the supplemental figure was to show this one IC-HHC pair that had matched RFLP but was ruled out based on SNP distance.

Critique 6. Overall phylogenetic lineages should be annotated to make clear if mutation rates are maybe lineage specific or if the observation is attributable to all MTBC strains (also in Figure 2).

Response:

We thank the reviewer for this suggestion. Using the program SNP-IT (Lipworth *et al.*, Emerging Infectious Diseases 2019, 25(3): 482–488), we confirmed that all clinical strains in this study belong to lineage 4. We have now included this information in the manuscript (results and methods). In the discussion, we have also added: “Finally, all the TB pairs in our study were infected with *M. tuberculosis* lineage 4. Although our study analysis benefits from this high degree of strain uniformity, it is possible that other TB lineages may exhibit different mutation rates x generation times during LTBI”. Overall, we believe that our study benefits substantially from the fact that all of the isolates are members of the same TB lineage, which is the most widely distributed lineage across the world.

Critique7. Line 352: Could it be that the expected genome size N was larger in the calculation than the actual size of the analyzed genome? Considering that the authors excluded repetitive elements from the SNP analysis (approximately 10% of the genome), as well as SNPs with insufficient coverage, the actual genome size might be less than 90%.

Response:

We agree that the actual genome size is smaller since we ignored genes in the PE PPE gene family. There are 165 PE PPE genes, spanned over 278,817 genomic positions (6.3% of the genome). Moreover, high coverage in our WGS data meant that only 2.7% of genomic sites had insufficient coverage (<20 reads at a genomic position), and the expected genome size was adjusted to 97.3% of reference genome length to account for this. However, if the coverage is lower, each of the estimated rates will be proportionally higher by the same amount since we used a constant coverage value for all pairs, and all of our inferences will be unchanged.

REVIEWER COMMENTS

Reviewer #1 (Remarks to the Author):

The revised version reflects the benefits of peer review: the authors have adequately addressed all substantive criticisms. Only the following minor errors should be corrected in preparing the final article:

Abstract “These results suggest that *M. tuberculosis* enters a quiescent state” Delete "into"

Intro (P4) “These conflicting studies illustrate how poorly the mutation rate, generation time and the accompanying physiological and metabolic status of *M. tuberculosis* ^are^ understood during LTBI.” Delete "is"

Results (P11) “to definitively draw conclusions” should be “to draw definitive conclusions”

Reviewer #2 (Remarks to the Author):

In brief, the authors find that subjects with LTBI and latency duration between 0-2 years have similar genetic distance (measured in SNPs) as those subjects with latency duration >2 years. They conclude that individuals who are latently infected for 0-2 years acquire similar numbers of SNPs as those are latently infected for >2 years.

The authors do a great job of rigorously excluding IC/HHC pairs that violate the critical assumption that the IC infected the HHC for each pair. Furthermore, the authors do a good job of analyzing the data in various ways (using two different SNP calling pipelines, excluding outliers, sensitivity analysis of the 14 TB pairs with the unique RFLP pattern, sequencing the 4 replicate H37Rv samples) to support the key finding.

They find that the # of SNPs between each IC/HHC pair does not correlate with the time elapsed between the active TB diagnosis of the IC and active TB diagnosis of the HHC case. From this, the authors conclude that the # of SNPs acquired during latency does not correlate with the duration of latency. While this conclusion is plausible, the authors then provide several analyses in an attempt to provide biological explanations based off of this limited data. I have a couple more revisions on these supporting analyses described below.

We thank the authors for responding to our suggestions, most of which we are satisfied with.

Critique 1: Mostly satisfied as changing binning from 5-6 years into 4-6 years is a big improvement, please see comment to Critique 9 below.

Critique 2: Satisfied

Critique 3: Satisfied

Critique 4: Satisfied

Critique 5: Satisfied

Critique 6: Mostly satisfied as changing binning from 5-6 years into 4-6 years is a big improvement, please see comment to Critique 9 below.

Critique 7: Satisfied

Critique 8: Satisfied

Critique 9: While breaking the data down into 3 analytic periods 0-2, 2-4 and 4-6 years is a marked improvement over the previous version of the MS (Figure 3A & 3B), I respectfully disagree that the authors that this harmonizes the MS. Both the Clinical characteristics (Table 1) and Oxidative stress during latency (Figure 5) in the results section analyze 2 analytic periods 0-2 years and > 2 years.

- If the main hypothesis is that mutation rate is higher from 0-2 yrs latency than >2 yrs latency why break up the latency duration for >2 yrs into 2-4 yrs & 4-6 yrs at all? Are the results for Figure 3 robust to pooling these two latter latency duration categories together? I suspect we would not observe as marked a decrease in the mutation rate for a pooled >2yrs category as is suggested by the last panel in Figure 3B for the (4-6] years latency duration, which is still based on a small sample size of (n=4).

- o This is important – it's one thing to observe that the #SNPs (and consequently mutation rate) between each IC/HHC pair does not appear to correlate with duration of latency. This is what I believe the data in this study shows clearly.

- o However, it is a bit of a leap to suggest that the mutation rate significantly decreases with duration of latency based off of this data alone.

- ▣ Breaking up the latency duration into 0-2 yrs, 2-4 yrs & 4-6 yrs illustrates the idea that mutation rate decreases with an increase in duration of latency but at the cost of having smaller sample sizes and making an arbitrary decision to split duration 2-4 & 4-6 years to support this hypothesis.

☐ From the discussion - “our results predicted that mutation rates decreased 7-fold when comparing ≤ 2 years of latency to >4 to 6 year-long LTBI”. I’m not convinced this data & analyses supports a significantly lower mutation rate (or a significantly longer generation time) during longer duration of latency and even the authors suggest that this may not be the case.

- I suggest either re-writing the results & discussion centered on this analysis to get rid of any strong claims on mutation rates/generation times based of this data alone or if you’re aiming to draw conclusions about the mutation rate then at least combining data points for 0-2 yrs & >2 yrs to keep in line with the rest of the text and avoid the suggestion of a lower mutation rate based off of a very small sample size of $n = 4$.

Critique 10 (added): In Figure 4 each point is another IC/HHC pair. The points should *generally* follow the trend in Figure 3A (decreasing as moving along the x-axis), since # SNPs between each pair is used in calculating the mutation rate (holding generation time fixed at 18 hours). While the pairs seem to agree (can be mapped between figures) between most points, there seems to be some disagreement from months 0-30.

- The outlier pair (point) occurs ~ 24 -30 months in Figure 3A but at 10-15 months in Figure 4.

- o As time increases along the x-axis in Figure 4, mutation rate decreases according to the formula, so I can see the point that corresponds to the outlier in Figure 3A in Figure 4 (CA3048/CA3078-3) at ~ 24 -30 months (point at roughly $3.0e-09$).

- o However, I don’t see which IC/HHC pair the outlier in Figure 4 at ~ 10 -15 months (point at roughly $1.7e-08$) could map to in Figure 3A, there are no outliers in this region (no pairs that have an usually large # SNPs/pair during this latency period). Can the authors explain this discrepancy? This could be a problem with one data point or a bug in the script affecting other data points.

Additional Minor Comments

- Table 1 - There are 24 IC/HHC pairs, and according to Figure 3 they are broken up as [0, 2] years ($n=11$); (2, 4] years ($n=9$); (4, 6] years ($n=4$). However according to Table 1, (0-2 years) has $n=11$ pairs and (>2 years) has $n=14$ pairs for a total of 25 pairs. Was 1 pair in the latter category counted twice?

- Figure 3A & Figure 4 – it would be appropriate if these figures had the same x-tick labels for comparing across figures.

- Figure 3B could use a dashed vertical line from 18 hours up toward and intersecting the grey dashed line to make the comparison of mutation rate (holding generation time constant at 18 hours) between the different latency periods clearer.

- Supplementary Methods – Modeling new SNP incidence during progression of LTBI

- o Why did the authors use the # of SNPs called from the SNPTB pipeline for this part of the analyses when it seems they used the SNPs called from the MTBseq pipeline for every other analysis?

- Methods – SNP identification

- o Looks like there is a typo “and high-confidence SNPs (Q score > 200)” should probably be “and high-confidence SNPs (Q score > 20)”

Reviewer #3 (Remarks to the Author):

The authors aimed to assess the mutation rate of MTB strains during latency, a period in which patients are asymptomatic and not contagious. Latent patients are assumed to be a large reservoir (there are estimates that 1/3 of the world population is infected; or was at least in contact with the bacteria) for new infections, once they develop active TB during their lifetime. Here, the authors compared SNP differences between index patients and their household contacts, who developed TB up to 5-6 years later, and analyzed if the pairwise distance increased with increased latency period, which it did not. This suggests either a lower mutation rate or an extended generation time (quiescent stage) the longer the bacteria are under control of the immune system. The authors conclude that this results in a reduced risk to acquire drug resistance mediating mutations, e.g. by reducing the population diversity in the latently infected patient, which has implications for preventive therapy as well.

This is a thorough and extensive revision, and I think the authors addressed my previous points and critics from other reviewers exhaustively. Due to the nature of latent TB infections (no culture) and low mutation rate per se, this is a difficult research question of course. But the authors precisely explain their inclusion and exclusion criteria, did some additional sensitivity analysis by removing outliers and reproduced their finding with another bioinformatics pipeline. It was a pleasure to read the revised version and I am only left with one question.

Once the latent patient develop active TB, I assume this goes a long with a “resurrection” of the bacteria. Maybe the authors can address this in a follow up paper, if the mutation rate goes back to normal while investigating sequential isolates from the HHC.

Overall: We thank the reviewers for their many positive comments about our revised manuscript including (reviewer #1) “The revised version reflects the benefits of peer review: the authors have adequately addressed all substantive criticisms”; (reviewer #2) “The authors do a great job of rigorously excluding IC/HHC pairs that violate the critical assumption that the IC infected the HHC for each pair. Furthermore, the authors do a good job of analyzing the data in various ways (using two different SNP calling pipelines, excluding outliers, sensitivity analysis of the 14 TB pairs with the unique RFLP pattern, sequencing the 4 replicate H37Rv samples) to support the key finding”; and (reviewer #3) “This is a thorough and extensive revision, and I think the authors addressed my previous points and critics from other reviewers exhaustively”.

Below, we address the final comments the reviewers.

REVIEWER #1

Critique: Only the following minor errors should be corrected in preparing the final article:

Abstract “These results suggest that *M. tuberculosis* enters a quiescent state” Delete “into”

Response: This has been fixed.

Critique: Intro (P4) “These conflicting studies illustrate how poorly the mutation rate, generation time and the accompanying physiological and metabolic status of *M. tuberculosis* ^are^ understood during LTBI.” Delete “is”

Response: This has been fixed.

Critique: Results (P11) “to definitively draw conclusions” should be “to draw definitive conclusions”

Response: This has been fixed.

REVIEWER #2

Critique: While breaking the data down into 3 analytic periods 0-2, 2-4 and 4-6 years is a marked improvement over the previous version of the MS (Figure 3A & 3B), I respectfully disagree that the authors that this harmonizes the MS. Both the Clinical characteristics (Table 1) and Oxidative stress during latency (Figure 5) in the results section analyze 2 analytic periods 0-2 years and > 2 years.

- If the main hypothesis is that mutation rate is higher from 0-2 yrs latency than >2 yrs latency why break up the latency duration for >2 yrs into 2-4 yrs & 4-6 yrs at all? Are the results for Figure 3 robust to pooling these two latter latency duration categories together? I suspect we would not observe as marked a decrease in the mutation rate for a pooled >2yrs category as is suggested by the last panel in Figure 3B for the (4-6] years latency duration, which is still based on a small sample size of (n=4).

Response: We would like to emphasize here that the Figure 4 in the earlier version treated duration of latency as a continuous variable (data was not binned) and fitting the mutation rate data to a Poisson model in this figure showed that the mutation rate declined significantly as a function of latency duration. We have now moved this result to Figure 3B, and have highlighted this result in the abstract.

We have now also placed a new analysis that categorized the latency periods into two analytic periods [0-2] and (2-6) years to show the change in the mutation rate over these time periods (new Figure 4). However, the mutation rate decline between these two intervals is not significant and has been mentioned as such in the manuscript. The three bin analysis remains significant and has been moved to the supplement for those readers who wish to see an alternative analysis that is more analogous to the approach shown in Fig 3B where we treated latency as a continuous variable

Critique: This is important – it's one thing to observe that the #SNPs (and consequently mutation rate) between each IC/HHC pair does not appear to correlate with duration of latency. This is what I believe the data in this study shows clearly. However, it is a bit of a leap to suggest that the mutation rate significantly decreases with duration of latency based off of this data alone.

Response: We have tried to make it clear that the observed results can be attributed to low mutation rates or long generation time during for longer latency periods. SNP incidence in a single generation (or another constant time period) is how the mutation rate is defined and calculated. However, generation times can vary as well and can affect the interpretation of the mutation rate, as discussed in the manuscript. Thus, we respectfully disagree with the reviewer that our conclusion is a bit of a leap. Assuming a constant generation time of 18h, the new Figure 3B clearly shows that the decline in mutation rate is statistically significant while considering latency duration as a continuous variable.

Critique: From the discussion - "our results predicted that mutation rates decreased 7-fold when comparing ≤ 2 years of latency to >4 to 6 year-long LTBI". I'm not convinced this data & analyses supports a significantly lower mutation rate (or a significantly longer generation time) during longer duration of latency and even the authors suggest that this may not be the case.

- I suggest either re-writing the results & discussion centered on this analysis to get rid of any strong claims on mutation rates/generation times based of this data alone or if you're aiming to draw conclusions about the mutation rate then at least combining data points for 0-2 yrs & >2 yrs to keep in line with the rest of the text and avoid the suggestion of a lower mutation rate based off of a very small sample size of $n = 4$.

Response: We have divided the data into the two periods as requested.

Critique: In Figure 4 each point is another IC/HHC pair. The points should *generally* follow the trend in Figure 3A (decreasing as moving along the x-axis), since # SNPs

between each pair is used in calculating the mutation rate (holding generation time fixed at 18 hours). While the pairs seem to agree (can be mapped between figures) between most points, there seems to be some disagreement from months 0-30.

- The outlier pair (point) occurs ~ 24-30 months in Figure 3A but at 10-15 months in Figure 4. As time increases along the x-axis in Figure 4, mutation rate decreases according to the formula, so I can see the point that corresponds to the outlier in Figure 3A in Figure 4 (CA3048/CA3078-3) at ~ 24-30 months (point at roughly $3.0e-09$). However, I don't see which IC/HHC pair the outlier in Figure 4 at ~10-15 months (point at roughly $1.7e-08$) could map to in Figure 3A, there are no outliers in this region (no pairs that have an usually large # SNPs/pair during this latency period). Can the authors explain this discrepancy? This could be a problem with one data point or a bug in the script affecting other data points.

Response: We thank the reviewer for carefully comparing the plots and identifying an issue in the figure, which is now corrected. There was indeed a bug in the script that plotted the raw data. In the new Figure 4 the outlier is a pair with a SNP difference of 4 with a latency period of 1. The outlier from Figure 3A occurs at 26 months and in Figure 4 is shown above the estimated curve; once normalized by the latency period this point appears relatively less extreme. The observed mutation rate data is now provided in the supplementary table S2.

Critique: Additional Minor Comments Table 1 - There are 24 IC/HHC pairs, and according to Figure 3 they are broken up as [0, 2] years (n=11); (2, 4] years (n=9); (4, 6] years (n=4). However according to Table 1, (0-2 years) has n=11 pairs and (>2 years) has n=14 pairs for a total of 25 pairs. Was 1 pair in the latter category counted twice?

Response: We thank the reviewer for catching this error. We have checked the data and there are 13 pairs for the interval >2 years. We have now fixed this typo.

Critique: Figure 3A & Figure 4 – it would be appropriate if these figures had the same x-tick labels for comparing across figures.

Response: Thank you for this suggestion. The tick mark placement are now harmonized between these two Figures.

Critique: Figure 3B could use a dashed vertical line from 18 hours up toward and intersecting the grey dashed line to make the comparison of mutation rate (holding generation time constant at 18 hours) between the different latency periods clearer.

Response: Thank you for this suggestion. A vertical line has been added at 18 hours.

Critique: Supplementary Methods – Modeling new SNP incidence during progression of LTBI Why did the authors use the # of SNPs called from the SNPTB pipeline for this part of the analyses when it seems they used the SNPs called from the MTBseq pipeline for every other analysis?

Response: We used the SNPTB analyses for the simulation because this analysis had non-zero values for SNP differences for all IC-HHC pairs. Since “average number of SNPs” was a starting parameter in the simulation, zero and low SNP differences values from the MTBSeq pipeline yielded an average mutation rate of 1.75, as opposed to 14.5 SNPs from the SNPTB pipeline. We thus chose the SNPTB value because it allows comparing models at a higher resolution.

Critique: Methods – SNP identification. Looks like there is a typo “and high-confidence SNPs (Q score > 200)” should probably be “and high-confidence SNPs (Q score > 20)”

Response: The Q score refers to the QUAL column in the VCF files, and the threshold for detecting high quality SNPs was 200. Thus, this is not a typo. We have updated the manuscript to clarify that this is QUAL score.

REVIEWER #3

Critique: This is a thorough and extensive revision, and I think the authors addressed my previous points and critics from other reviewers exhaustively. Due to the nature of latent TB infections (no culture) and low mutation rate per se, this is a difficult research question of course. But the authors precisely explain their inclusion and exclusion criteria, did some additional sensitivity analysis by removing outliers and reproduced their finding with another bioinformatics pipeline. It was a pleasure to read the revised version and I am only left with one question. Once the latent patient develop active TB, I assume this goes a long with a “resurrection” of the bacteria. Maybe the authors can address this in a follow up paper, if the mutation rate goes back to normal while investigating sequential isolates from the HHC.

Response: This is indeed an interesting question and we agree with the reviewer that it should be addressed in a future study. As our work in the most recent version of our manuscript shows, the consensus sequence of a “revived” in vitro culture of H37Rv had undetectable mutation rates under standard culture and sequencing conditions. However, this rate could be somewhat higher during human infections, a possibility that is considered in one of the four models that we describe in Fig S9 in the supplement.

REVIEWERS' COMMENTS:

Reviewer #2 (Remarks to the Author):

I commend the authors on the edits and on combining the two later time periods. Overall the edits are satisfactory, but I'm still not clear how the Poisson regression was done. Their premise as stated in the paper's title of different evolutionary rates in latency hinges on this modeling, as it's clear from Figure 3A that there is not a measurable decrease in SNP count over time grossly without making assumption about fixed generation time. At the moment the description of how this Poisson regression was built is only detailed to some extent in Figure 3 legend:

"Mutation rate as a function of latency duration (months between IC and HHC 607 TB diagnosis). The generation time is held constant at 18 hours as seen in actively replicating *M. tuberculosis* in vitro. The smooth line shows the Poisson regression fit to the number of SNPs using t , the observed latency period, as the dependent variable for each participant using an offset $N*t/g$, where N is $.973*$ genome size and t/g is the number of generations and g is set to 18 hours. The coefficient for t was significantly different from 0.0 (two-sided $p < 0.001$)"

For example if this is a Poisson rate model, why is the coefficient described as the coefficient for t ? is it not the coefficient of SNPs observed? also the coefficient is not actually given, is it a very small negative value?

Also it makes me a little nervous that the two SNP calling methods they used reported vastly different SNP counts ("average mutation rate of 1.75 vs 14.5"), is it because a larger portion of the genome was masked with MTBseq? At the very least they should state that these two methods disagreed, and give a plot of SNPs vs time using the MTBseq pipeline.

REVIEWERS' COMMENTS:

Reviewer #2 (Remarks to the Author):

I commend the authors on the edits and on combining the two later time periods. Overall the edits are satisfactory, but I'm still not clear how the Poisson regression was done. Their premise as stated in the paper's title of different evolutionary rates in latency hinges on this modeling, as it's clear from Figure 3A that there is not a measurable decrease in SNP count over time grossly without making assumption about fixed generation time. At the moment the description of how this Poisson regression was built is only detailed to some extent in Figure 3 legend:

"Mutation rate as a function of latency duration (months between IC and HHC 607 TB diagnosis). The generation time is held constant at 18 hours as seen in actively replicating *M. tuberculosis* in vitro. The smooth line shows the Poisson regression fit to the number of SNPs using t , the observed latency period, as the dependent variable for each participant using an offset $N*t/g$, where N is $.973*$ genome size and t/g is the number of generations and g is set to 18 hours. The coefficient for t was significantly different from 0.0 (two-sided $p < 0.001$)"

For example if this is a Poisson rate model, why is the coefficient described as the coefficient for t ? is it not the coefficient of SNPs observed? also the coefficient is not actually given, is it a very small negative value?

Response:

There was a typo in the text (now corrected) that made the model used unclear; t is the **independent** variable in the regression and SNP difference is the dependent variable. By using the offset, the parameters are used to estimate the mutation rate per bp/generation as a function of latency. The coefficient for t (in months) as the reviewer notes was a negative value, it was -0.053 (95% confidence interval: -0.078, -0.029). We felt that the plot more meaningfully displayed the results which uses both the intercept and coefficient estimates but now provide the point estimate for the coefficient in the footnote.

Also it makes me a little nervous that the two SNP calling methods they used reported vastly different SNP counts ("average mutation rate of 1.75 vs 14.5"), is it because a larger portion of the genome was masked with MTBseq? At the very least they should state that these two methods disagreed, and give a plot of SNPs vs time using the MTBseq pipeline.

Response:

The MTBseq pipeline identified fewer SNPs than the SNPTB pipeline due to more restrictive SNP filtering not because of genome masking. The plot of SNPs vs time using SNP data generated by the MTBseq pipeline requested by the reviewer is shown in Figure 3A, and the data is listed in Table S2. It is clear from

the figure that the trends are the same as that shown for the SNPTB pipeline in Figure S5 (data in Table S3), except that all of the SNP numbers are higher.

The section “SNP identification” in Methods discussed the SNP filtering rules followed by these methods. To reiterate:

SNP filtering by MTBseq: SNPs were ignored if

1. Less than 75% of the reads reported the SNP, including at least 4 reads in both forward and reverse direction.
2. At least 4 reads had a minimum Phred score of 20.
3. SNP belonged to repetitive regions.
4. SNP belonged to drug resistance genes.
5. SNPs that were within a window of 12 bases.

SNP filtering by SNPTB: SNPs were ignored if:

1. They had a QUAL score < 200. This score is computed by SAMtools based on a lot of factors and outputs the phred-scaled quality score for the SNP, with higher scores indicating high confidence in the SNP call. The QUAL score ranges from 2 to 225, and scores > 200 are considered high quality.
2. SNP belonged to the repetitive regions in the genome (PE/PPE genes).

Thus, there is not a direct comparison of the filtering between the two methods. SAMtools does detect some SNPs with high confidence (QUAL score > 200) even if the SNP is in ~50% of the reads. On the other hand, MTBseq removes SNPs in drug resistance genes and SNPs that are close to each other, which SNPTB does not.

Thus, MTBseq filtering can be seen as more stringent than SNPTB's. We have now highlighted this observation in the legend of Supplementary Figure S5 that shows the SNP vs time data for SNPTB.